# NUMERICAL PITFALLS IN POLICY GRADIENT UPDATES

## ABSTRACT

Numerical instability, such as gradient explosion, is a fundamental problem in practical deep reinforcement learning (DRL) algorithms. Beyond anecdotal debugging heuristics, there is a lack of systematic understanding of the causes for numerical sensitivity that leads to exploding gradient failures in practice. In this work, we demonstrate that the issue arises from the ill-conditioned density ratio in the surrogate objective that comes from importance sampling, which can take excessively large values during training. Perhaps surprisingly, while various policy optimization methods such as TRPO and PPO prevent excessively large policy updates, their optimization constraints on KL divergence and probability ratio cannot guarantee numerical stability. This also explains why gradient explosion often occurs during DRL training, even with code-level optimizations. To address this issue, we propose the Vanilla Policy Gradient with Clipping algorithm, which replaces the importance sampling ratio with its logarithm. This approach effectively prevents gradient explosion while achieving performance comparable to PPO.

## 1 INTRODUCTION

Deep reinforcement learning (DRL) has demonstrated its effectiveness in various domains (Mnih et al., 2015; Silver et al., 2017; Vinyals et al., 2019; Ouyang et al., 2022). Despite these successes, the reliability and numerical stability of DRL algorithms remain fundamental limitations that hinder their application in real-world environments. In particular, DRL algorithms are often observed to have *numerical instability* issues, such as exhibiting the major problem of gradient explosion that destroys the learning progress. Existing work on the brittleness of DRL methods attributes this to either the high variance in actions (Fujita & Maeda, 2018), the gradient estimator (Liu et al., 2020), or the sensitivity to hyperparameters (Henderson et al., 2018).

In this paper, we perform a rigorous analysis of the root cause of numerical instability in policy optimization algorithms such as Trust Region Policy Optimization (TRPO) and Proximal Policy Optimization (PPO) (Schulman et al., 2015a; 2017). Gradient explosion is a consequence of overflows caused by computations involving excessively large numbers that exceed the limits of floating-point arithmetic. We show that the long horizons and large value variance are not the core reason for overflow, especially under action clipping as commonly used in standard DRL implementations (Duan et al., 2016; Raffin et al., 2021; Bou et al., 2024). Another popular explanation is that the covariance matrix of the stochastic policy can become singular during training, leading to gradient explosion. However, our experiments in Appendix B show that arithmetic overflows can occur even when the standard deviations are lower-bounded. Instead, our analysis indicates that the root cause is the importance sampling steps in TRPO and PPO (Schulman et al., 2015a; 2017), where the probability ratio can take exponentially large values, especially in the case of Gaussian distributions. *Specifically, we analyze the condition number of the probability ratio, and demonstrate that the probability ratio is exponential with respect to a line integral in the policy space, which can lead to arithmetic overflows and cause gradient explosion.*

In particular, we demonstrate that optimization constraints proposed in TRPO/PPO, such as limiting the KL-divergence and clipping, do not prevent numerical instability. We show that the probability ratio in importance sampling can grow exponentially fast, ultimately causing arithmetic overflows, even while the KL divergence remains small. This indicates that various policy optimization methods that prevent excessively large policy updates do not prevent numerical problems. Indeed, the probability ratios are not guaranteed to stay small, especially when the policy network's output is

further clipped in the action space, leading to excessively small values in the probability density functions and large condition numbers in the density ratio.

Given that importance sampling is the root cause of the exploding gradient issue, we propose an algorithm called Vanilla Policy Gradient with Clipping, a modified version of the PPO algorithm in which the importance sampling ratio is replaced with its logarithm. We conduct experiments on continuous-control benchmarks to evaluate its effectiveness. Specifically, we find that this algorithm performs comparably to PPO on small policy networks where no gradient explosion occurs and maintains good performance on large policy networks where PPO encounters numerical issues. Additionally, the influence of other code-level techniques, such as reward scaling and learning rates, is discussed in Appendix B.

Overall, our findings suggest that the TRPO/PPO loss is inherently ill-conditioned due to importance sampling and action clipping, which directly contribute to the numerical instability of deep policy gradient methods. This issue should not be viewed merely as a hyperparameter tuning problem but as a fundamental limitation of DRL algorithms that requires more attention.

## 2 RELATED WORK

**Deep policy gradients in practice.**    It has been reported that implementation details and code-level optimizations fundamentally impact the performance of deep policy gradient algorithms such as TRPO and PPO (Henderson et al., 2018; Engstrom et al., 2020; Andrychowicz et al., 2021). In particular, empirical studies have shown that the most significant discrepancies between theory and implementation arise in gradient estimation, value prediction, and optimization landscapes (Ilyas et al., 2020). These findings were later supported by theoretical studies, which indicated that they stem from the fractal structures present in both value and policy landscapes (Wang et al., 2023). Our work contributes to this line of research by providing an in-depth study on the numerical stability of deep policy gradient methods.

**Exploding gradients in deep learning.**    Many neural network architectures have been reported to suffer from the gradient explosion issue, including deep multi-layer networks (Bengio et al., 1994; Glorot & Bengio, 2010) and residual neural networks (RNNs) (Pascanu et al., 2013). This problem is closely linked to the depth of the network architecture (Philipp et al., 2017; Schoenholz et al., 2017), which can result in a long chain of Jacobian multiplications. In the context of reinforcement learning (RL), Liu et al. (2018) points out that the product of density ratios over a long horizon can grow exponentially, leading to exploding variance in off-policy estimation. Also, Khorasani et al. (2023) proposes to use second-order information in the form of Hessian vector products to bypass importance sampling weights in policy gradient. In this work, we demonstrate that even a single density ratio can take excessively large values and cause numerical overflow, and propose an algorithm that drops the importance sampling from PPO objective while achieving comparable performance. Wang et al. (2024) investigates gradient explosion in the context of generative adversarial imitation learning (GAIL), suggesting that a large reward function can cause exploding gradients. However, our empirical results in Section B indicate that this does not fully explain gradient explosion in policy gradient algorithms, as scaling down the reward does not completely overcome the issue.

## 3 NUMERICAL INSTABILITY CAUSED BY IMPORTANCE SAMPLING

**TRPO and PPO objective.**    The original policy gradient estimator has the form

$$\nabla \mathcal{J}(\theta) = \mathbb{E}_{(s_t, a_t) \sim \pi_\theta} \Big[ \nabla_\theta \log \pi_\theta(a_t|s_t) \, Q^\pi(s_t, a_t) \Big], \tag{1}$$

where $Q^\pi$ is the $Q$-function of the current policy $\pi_\theta$ (Sutton et al., 1999). One difficulty of directly optimizing equation 1 is due to the complex dependency of sampled data on $\pi_\theta$ (Schulman et al., 2015a). To address it, TRPO proposes a surrogate objective that incorporates importance sampling:

$$\max_\theta \quad \hat{\mathbb{E}}_{(s_t, a_t) \sim \pi} \Big[ \frac{\pi_\theta(a_t|s_t)}{\pi(a_t|s_t)} \hat{A}^\pi(s_t, a_t) \Big], \tag{2}$$

where $\hat{\mathbb{E}}_{(s_t, a_t) \sim \pi}[\cdot]$ denotes the estimated expectation over sampled trajectories and $\hat{A}^\pi(s_t, a_t)$ the estimated advantage function at $(s_t, a_t)$ which is usually obtained through Generalized Advantage

Estimation (GAE, Schulman et al. (2015b)). Let $L(\theta)$ denote the surrogate objective in equation 2. It has been proved that the gradients of equation 1 and equation 2 coincide at $\pi = \pi_\theta$. Additionally, both TRPO and PPO suggest that the current policy $\pi_\theta$ should not deviate too much from the old policy $\pi$, leading to various optimization constraints which will be discussed in Section 5.

**Condition number and numerical stability.** We now briefly introduce the concept of the condition number from numerical analysis, which measures how much the output of an algorithm changes in response to a small change in its input:

**Definition 3.1.** *(Condition number) Given a function $f$ and an input $x$, let $\delta x$ be the error in $x$ and $\delta f(x) = f(x + \delta x) - f(x)$ be the corresponding error resulted in the output. We define the absolute condition number of $f$ at $x$ as*

$$\kappa = \lim_{\delta \to 0} \sup_{|\delta x| \le \delta} \frac{|\delta f(x)|}{|\delta x|}. \tag{3}$$

*In particular, when the function $f$ is differentiable, it further has*

$$\kappa = \|J_f(x)\|,$$

*where $J_f(x)$ is the Jacobian of $f$ at $x$.*

A detailed study can be found in Trefethen & Bau (1997). The following concept of machine precision defines the threshold of numerical overflows:

**Definition 3.2.** *(Machine precision) Let $e > 0$ denote the machine precision throughout this paper, any quantity with its absolute value smaller than $e$ will be considered zero.*

**Importance sampling with clipped actions.** Consider the probability ratio $p_\theta(a|s) = \frac{\pi_\theta(a|s)}{\pi(a|s)}$ in the loss function. This ratio might become ill-conditioned when $\pi(a|s)$ is very small. However, it is rarely reported in practice that importance sampling itself causes any numerical issues. This is because, when importance sampling is applied to modify a probabilistic distribution, the probability ratio $p_\theta(a|s)$ is only evaluated at data points sampled from the distribution $\pi(\cdot|s)$. This means that the probability of obtaining an action $a$ where $\pi(a|s) \ll 1$ is also very small, thereby rarely causing numerical instability. For example, let $a \sim \mathcal{N}(\mu_0(s), \Sigma_0)$ be $m$-dimensional Gaussian random variable with probability density function $\pi(a|s)$, then we have

$$P(\pi(a|s) < e) \le 2\sqrt{2\pi}m(e\sqrt{\det \Sigma_0})^{\frac{1}{m}}, \tag{4}$$

where $e \ll 1$ is the machine precision.

Although it is unlikely for $\pi(a|s)$ to take extremely small values if $a$ is directly sampled from $\pi$ itself, as shown in equation 4, it is important to note that the actual action space usually has a bounded range due to physical or environmental constraints. Therefore, a common practice in DRL is to clip the sampled actions before feeding them into the simulator. Without loss of generality, we assume that the action space is of the box form $\mathcal{A} = [-\beta, \beta]^m$ for some positive $\beta$, and write the action-clipping operation as $\phi : \mathbb{R}^m \to [-\beta, \beta]^m$. Namely, for any $a = (a_1, ..., a_m) \in \mathbb{R}^m$, $\phi(a) = (a'_1, ..., a'_m)$ where $a'_i = \max(-\beta, \min(a_i, \beta))$.

Note that the probability ratio is evaluated *after* action clipping (Raffin et al., 2021).That is, $p_\theta(\phi(a)|s)$ is used instead of $p_\theta(a|s)$ when computing the objective in equation 2. Consequently, the distance between the mean $\mu(s)$ and the clipped action $\phi(a)$ is no longer guaranteed to be small with high probability as suggested in equation 4, especially when $\mu(s)$ lies outside the action space $\mathcal{A}$:

**Theorem 3.1.** *Let $a \sim \mathcal{N}(\mu(s), \Sigma)$ be $m$-dimensional Gaussian random variable with probability density function $\pi(\cdot|s)$, and $\mathcal{A} = [-\beta, \beta]^m$ with $\beta > 0$. Let $\phi : \mathbb{R}^m \to \mathcal{A}$ be the action-clipping transformation, then*

    *1. Suppose that $\mu(s) \in \mathcal{A}$, then for any $a \in \mathbb{R}^m$,*

$$P(\pi(\phi(a)|s) < e) \le 2\sqrt{2\pi}m(e\sqrt{\det \Sigma_0})^{\frac{1}{m}},$$

    *where $e$ is the machine precision;*

2. *If $\mu(s) \notin \mathcal{A}$, then for any $a \in \mathbb{R}^m$ it has*

$$\pi(\phi(a)|s) \leq \frac{1}{(2\pi)^{\frac{m}{2}} \sqrt{\det \Sigma}} \exp(-\frac{d^2}{2\lambda_{max}}),$$

*where $d = dist(\mu(s), \mathcal{A})$ be the distance between $\mu(s)$ and the action space $\mathcal{A}$, $\lambda_{max}$ is the largest eigenvalue of $\Sigma$.*

Therefore, $\pi(\phi(a)|s)$ may fall below machine precision $e$ when $\mu(s)$ is far from the action space $\mathcal{A}$ or when the standard deviations of $\pi(\cdot|s)$ are sufficiently small. In either case, $\pi(\phi(a)|s)$ becomes equal to $0$ on the machine, and dividing by it can cause numerical issues.

**Tanh-Gaussian transformation.** It is worth mentioning that we can also bound the action by applying an invertible squashing function, such as $\tanh$, to the Gaussian samples (Haarnoja et al., 2018). However, this approach still suffers from the ill-conditioned nature of importance sampling. For instance, consider the modified probability density function in the one-dimensional case, given by

$$\pi'(a|s) = \pi(u|s)(1 - \tanh^2(u))^{-1} \tag{5}$$

where $u \sim \pi$ is a Gaussian random variable and $a = \tanh(u)$. Note that when the mean of $\pi$ is too large, it is likely that the random variable $u$ will take on extreme values, resulting in an excessively small value for the term $1 - \tanh^2(u)$ and causing numerical overflow in subsequent steps.

**Ill-conditioned probability ratios.** We have seen that the action-clipping transformation in deep policy gradients can significantly affect the numerical stability of importance sampling. However, one might argue that small values in $\pi(a|s)$ can be counterbalanced by small values in $\pi_\theta(a|s)$ assuming that the current policy $\pi_\theta$ is constrained to remain within a neighborhood of $\pi$. This is particularly true in the first optimization step of each epoch, where $\pi_\theta$ is reset to $\pi$ and all probability ratios are equal to $1$. We will show that when action clipping is applied, the condition number of the probability ratio can become very large, causing $p_\theta(\phi(a)|s)$ to grow exponentially fast and leading to instability as $\pi_\theta$ deviates from $\pi$. Let us again consider a Gaussian policy $\pi_\theta(\cdot|s) = \mathcal{N}(\mu(s;\theta), \Sigma(\theta))$, where $\mu(s)$ is typically parameterized by a neural network and $\Sigma(\theta)$ is positive-definite and state-independent. Let $\pi(\cdot|s) = \mathcal{N}(\mu(s;\theta_0), \Sigma(\theta_0)))$, the probability ratio is calculated as

$$p_\theta(\phi(a)|s) = \frac{\pi_\theta(\phi(a)|s)}{\pi(\phi(a)|s)}$$

$$= \sqrt{\frac{\det \Sigma(\theta_0)}{\det \Sigma(\theta)}} \frac{\exp\left(-\frac{1}{2}(\phi(a) - \mu(s;\theta))^T \Sigma(\theta)^{-1}(\phi(a) - \mu(s;\theta))\right)}{\exp\left(-\frac{1}{2}(\phi(a) - \mu(s;\theta_0))^T \Sigma(\theta_0)^{-1}(\phi(a) - \mu(s;\theta_0)))\right)}.$$

To better illustrate how the action-clipping transformation $\phi$ affects the condition number of $p_\theta$, we further assume that $\Sigma(\theta) \equiv constant$. The general case with parameterized standard deviations is discussed in Appendix C. Then, the gradient of the probability ratio is given by

$$\nabla_\theta \, p_\theta(\phi(a)|s) = p_\theta(\phi(a)|s)(\phi(a) - \mu(s;\theta))^T \Sigma(\theta)^{-1} \frac{\partial \mu(s;\theta)}{\partial \theta}, \tag{6}$$

where $\frac{\partial \mu(s;\theta)}{\partial \theta}$ is the Jacobian of $\mu$ with respect to its parameters. Note that equation 6 is equivalent to

$$\nabla_\theta \, \log\left(p_\theta(\phi(a)|s)\right) = (\phi(a) - \mu(s;\theta))^T \Sigma(\theta)^{-1} \frac{\partial \mu(s;\theta)}{\partial \theta}, \quad p_{\theta_0}(\phi(a)|s) = 1,$$

which describes the gradient flow of $p_\theta$ in the policy space. Let $\mathcal{C}$ be the trajectory of policy parameters generated by PPO in the policy space, starting at $\theta_0$ and ending at $\theta_1$. Applying the gradient theorem yields

$$p_{\theta_1}(\phi(a)|s) = \exp\left(\int_\mathcal{C} (\phi(a) - \mu(s;\theta))^T \Sigma(\theta)^{-1} \frac{\partial \mu(s;\theta)}{\partial \theta} \cdot d\mathbf{r}\right) \tag{7}$$

where the term in the exponent is the line integral along $\mathcal{C}$. Therefore, the condition number $\kappa_\theta = \|\nabla_\theta \, p_\theta(\phi(a)|s)\|$ can be extremely large as illustrated in Figure 1, implying that the probability ratio may overflow when

- The standard deviations of $\pi$ are small so that the matrix $\Sigma(\theta)^{-1}$ has large eigenvalues;

- The action is clipped and the mean $\mu(s)$ lies outside $\mathcal{A}$ so that $|\phi(a) - \mu(s;\theta)|$ is large (illustrated in Figure 1 (c));

- The policy parameterization $\mu(\cdot;\theta)$ is complex (e.g. large policy networks) so that a small change in the parameter can lead to a dramatic difference in the output.

Some DRL libraries further transform the standard deviations to prevent low variances. However, gradient explosion still occurs with those code-level techniques, indicating that the ill-conditioned probability ratio is the primary cause of the issue.

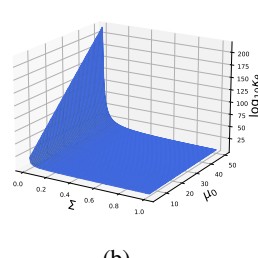 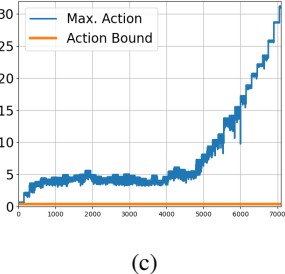

(a) (b) (c)

Figure 1: (a) When $\mu_0$ (the mean of $\pi_0$) is large, the condition number $\kappa_\theta$ of the probability ratio $p_\theta$ changes exponentially when $\mu$ (the mean of $\pi$) deviates from $\mu_0$; (b) $\kappa_\theta$ can become extremely large when the variance $\Sigma$ is small, particularly in cases of large means; (c) The action with the largest norm in each mini-batch grows outside the boundary during PPO training.

## 4 EMPIRICAL ANALYSIS OF GRADIENT EXPLOSION

In the previous section, we theoretically demonstrated the ill-conditioned nature of importance sampling in TRPO/PPO methods. In this section, we present experimental evidence to support the theory. The experimental setup follows that described in Appendix A.

**Gradient explosion is always accompanied by excessively large probability ratios.** We run the Humanoid-v4 environment on five random seeds, all of which fail due to arithmetic overflow. In Figure 2, we observe that the maximum of probability ratio $\max_t \log_{10}\left(p_\theta(a_t|s_t)\right)$ takes excessively large values across all five random seeds. A widely accepted explanation for gradient explosion is that the standard deviations of the Gaussian policy can become very small, leading to potential singularity issues in the covariance matrix $\Sigma$. While this can indeed cause numerical instability, it is not the only reason for gradient explosion. In Figure 6, we observe that the gradient explodes even before the standard deviations become small.

**Large mean causes gradient explosion.** We have mentioned that large outputs generated by the policy network are one of the primary causes of gradient explosion. To verify this analysis, we apply an additional transformation $g(\cdot)$ to the mean $\mu(s)$ to prevent it from leaving the action space $\mathcal{A}$. The new Gaussian policy is defined as $a \sim \mathcal{N}(g(\mu(s)), \Sigma(s))$, where $g(\mu(s)) \in \mathcal{A}$. In Table 1, we consider two transformations: the action-clipping function $g = \phi$, defined in Section 3, and the hyperbolic tangent activation function $g = \tanh$. We observe that both transformations effectively prevent the exploding gradient issue across three MuJoCo environments, indicating that having deviated means is indeed one of the main causes of numerical instability. Meanwhile, we also observe that the final returns are significantly lower than the benchmark (Raffin et al., 2021). This may be attributed to the fact that when the mean $\mu(s_t)$ goes outside $\mathcal{A}$ for some state $s_t$, the gradient norm

$$\nabla_\theta \, g(\mu(s_t;\theta)) = \left.\frac{\mathrm{d}g(\mu)}{\mathrm{d}\mu}\right|_{\mu=\mu(s_t)} \frac{\partial\mu(s_t;\theta)}{\partial\theta},$$

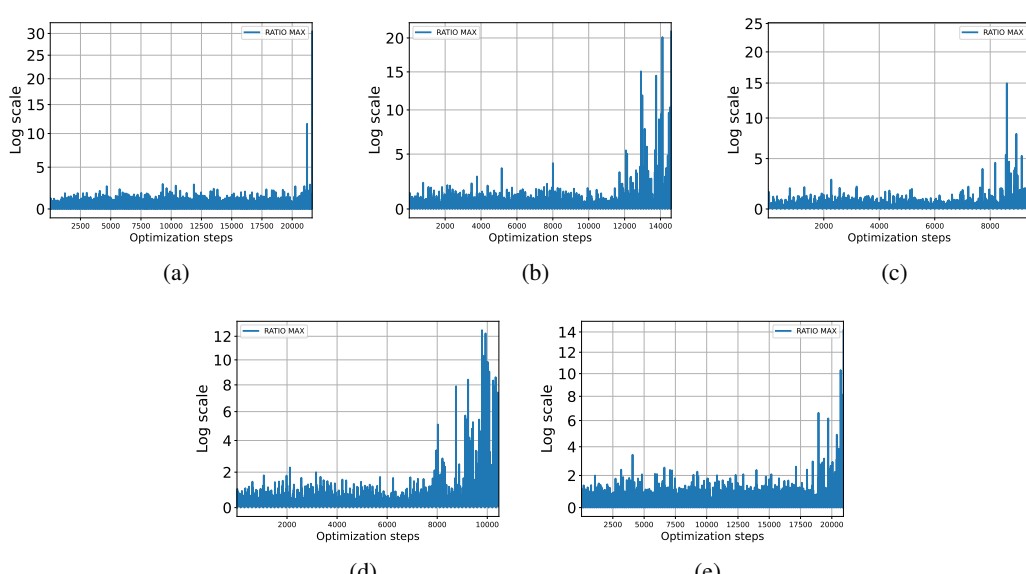

(a)        (b)        (c)

(d)        (e)

Figure 2: The maximum of probability ratio after clipping takes exponentially large values when the gradient explodes in five individual runs.

| ENVIRONMENT | TRANSFORMATION | FINAL RETURN | EXPLOSION RATE |
|---|---|---|---|
| HOPPER-V4 | CLIPPING | $332 \pm 129$ | 0% |
| HOPPER-V4 | tanh | $234 \pm 106$ | 0% |
| HOPPER-V4 | NONE | N/A | 100% |
| WALKER2D-V4 | CLIPPING | $692 \pm 346$ | 0% |
| WALKER2D-V4 | tanh | $434 \pm 60$ | 0% |
| WALKER2D-V4 | NONE | N/A | 100% |
| HUMANOID-V4 | CLIPPING | $586 \pm 102$ | 0% |
| HUMANOID-V4 | tanh | $530 \pm 72$ | 0% |
| HUMANOID-V4 | NONE | N/A | 100% |

Table 1: Performance of mean transformation via clipping function or $\tanh$ function in three MuJoCo environments. For Hopper-v4 and Walker2d-v4, we directly apply $\tanh$ as their action input range is between $-1$ and $1$. For the Humanoid-v4 task, we rescale the mapping to $g = 0.4 * \tanh$, as the action inputs are bounded between $-0.4$ and $0.4$.

becomes exponentially small or even equal to zero, depending on the transformation $g$. Notably, we may have $|\frac{\mathrm{d}g(\mu)}{\mathrm{d}\mu}|_2 \simeq 0$ due to the vanishing gradient property of the transformation $g$ when $\mu(s_t) \notin \mathcal{A}$. As a result, policy gradient algorithms are more likely to converge towards sub-optimal regions in this case.

# 5   OPTIMIZATION CONSTRAINTS CANNOT GUARANTEE NUMERICAL STABILITY

In previous sections, we demonstrated that importance sampling can be numerically unstable when the probability ratio is evaluated at transformed sample points. This is particularly problematic in deep policy gradient methods like TRPO and PPO, which incorporate the probability ratio into their surrogate objective functions. This integration is the fundamental reason for the numerical instability of these algorithms. However, it should be noted that both TRPO and PPO also apply certain constraints to the original optimization problem, such as KL divergence and probability ratio clipping. In this section, we will explain why these optimization constraints are insufficient to resolve the numerical issues as they are supposed to.

**KL divergence.** In TRPO algorithm (Schulman et al., 2015a), it proposes to use the KL divergence between the current and old policy as a constraint to prevent large updates during policy improvement. Specifically, the KL divergence of two continuous probability distributions $P$ and $Q$ is defined as

$$D_{KL}(P \parallel Q) = \int_\Omega p(x) \, \log \frac{p(x)}{q(x)} \, dx, \tag{8}$$

where $p$ and $q$ are the probability density functions of $P$ and $Q$, respectively. When optimizing the objective equation 2, TRPO applies the following hard constraint to the KL distance between successive policies

$$\hat{\mathbb{E}}_{s \sim \pi} \Big[ D_{KL}(\pi_\theta(\cdot|s) \parallel \pi(\cdot|s)) \Big] \leq \delta, \tag{9}$$

where $\delta > 0$ is a small quantity called the KL stepsize. Note that for many commonly used distributions, including Gaussian, the KL divergence between $\pi_\theta$ and $\pi$ can be analytically calculated via a closed-form formula. For instance, consider two multivariate Gaussian policies $\pi_\theta(\cdot|s) \sim \mathcal{N}(\mu(s), \Sigma_0)$ and $\pi(\cdot|s) \sim \mathcal{N}(\mu_0(s), \Sigma_0)$ with identical covariance matrices. The KL divergence is

$$D_{KL}(\pi_\theta(\cdot|s) \parallel \pi(\cdot|s)) = \frac{1}{2}(\mu(s) - \mu_0(s))^T \Sigma_0^{-1}(\mu(s) - \mu_0(s)). \tag{10}$$

While the above constraint can effectively prohibit $\pi_\theta$ from making large updates steps in the distribution space, it may not be able to prevent the probability ratio $p_\theta(a|s)$ from taking extremely large values at some state-action pair $(s, a)$ for two reasons: First, the KL divergence $D_{KL}(\pi_\theta(\cdot|s) \parallel \pi(\cdot|s))$ measures the averaged distance between two distributions over the entire space $\mathbb{R}^m$, while $p_\theta(a|s)$ is determined at a specific point $a \in \mathbb{R}^m$, which means that we can find some $a \in \mathbb{R}^m$ such that the probability ratio $p_\theta(a|s)$ takes a large value even if the KL distance $D_{KL}(\pi_\theta(\cdot|s)$ is small. Second, the absolute value of KL divergence $|D_{KL}(\pi_\theta(\cdot|s) \parallel \pi(\cdot|s))|$ is quadratic with respect to the policy update $|\mu(s) - \mu_0(s)|$, while the probability ratio $\pi_\theta(a|s)$ grows exponentially with $|\mu(s) - \mu_0(s)|$ especially for excessively small $\pi(a|s)$.

**Probability ratio clipping.** While TRPO involves second-order computations that are usually expensive in practice, PPO (Schulman et al., 2017) proposes a clipped surrogate objective

$$L^{CLIP}(\theta) = \hat{\mathbb{E}}_{(s_t, a_t) \sim \pi} \Big[ \min \Big( p_\theta \hat{A}^\pi(s_t, a_t), \text{clip}(p_\theta, 1 - \epsilon, 1 + \epsilon) \hat{A}^\pi(s_t, a_t)) \Big) \Big] \tag{11}$$

where $\epsilon \in (0, 1)$ is the clipping parameter. The clipped loss equation 11 is expected to block large updates in the policy that can improve the objective (i.e., $\hat{A}^\pi(s_t, a_t) > 0$), and allow them if they make the objective worse ($\hat{A}^\pi(s_t, a_t) < 0$). This property, however, makes it possible that some ill-conditioned probability ratios are evaluated when computing $L^{CLIP}(\theta)$.

In particular, consider a given state-action pair $(s, a)$ and the estimated advantage $\hat{A}^\pi(s, a)$:

- If $\hat{A}^\pi(s, a) \geq 0$, it has

$$\min \Big( p_\theta \hat{A}^\pi(s, a), \text{clip}(p_\theta, 1 - \epsilon, 1 + \epsilon) \hat{A}^\pi(s, a)) \Big) = (1 + \epsilon) \hat{A}^\pi(s, a),$$

  whenever $p_\theta \geq 1 + \epsilon$. Thus, it automatically blocks extremely large values in $p_\theta$ when evaluating $L^{CLIP}(\theta)$;

- If $\hat{A}^\pi(s, a) < 0$, it is *possible* that the machine directly evaluates large probability ratios, since

$$\text{Integrand in equation 11} = \min \Big( p_\theta \hat{A}^\pi(s, a), \text{clip}(p_\theta, 1 - \epsilon, 1 + \epsilon) \hat{A}^\pi(s, a)) \Big)$$

$$= \hat{A}^\pi(s, a) \max \Big( p_\theta, \text{clip}(p_\theta, 1 - \epsilon, 1 + \epsilon) \Big)$$

$$= \hat{A}^\pi(s, a) \max \Big( p_\theta, 1 - \epsilon \Big)$$

$$= p_\theta \, \hat{A}^\pi(s, a)$$

  when $p_\theta \geq 1 - \epsilon$, which then may lead to numerical overflows.

In practice, the expectation in equation 11 is estimated by the average over sampled trajectories

$$L^{CLIP}(\theta) \simeq \frac{1}{T} \sum_{t=0}^{T-1} \min\left(p_\theta \hat{A}^\pi(s_t, a_t), \text{clip}(p_\theta, 1-\epsilon, 1+\epsilon)\hat{A}^\pi(s_t, a_t))\right).$$

Therefore, numerical errors may occur if there exists any pair $(s_t, a_t)$ with negative advantage $\hat{A}^\pi(s_t, a_t) < 0$ and large probability ratio $p_\theta(a_t|s_t) \gg 1$.

**Empirical results.** We perform experiments to validate aforementioned theoretical analysis. In Figure 3 (a), we see that the KL divergence $D_{KL}(\pi_\theta(\cdot|s) \parallel \pi(\cdot|s))$ remains small even when the probability ratio $p_\theta(a|s)$ takes very large values, indicating that $D_{KL}(\pi_\theta(\cdot|s) \parallel \pi(\cdot|s))$ cannot ensure numerical stability. We also test the following KL-penalized objective

$$L^{KLPEN}(\theta) = \hat{\mathbb{E}}_t\left[p_\theta(a_t|s_t)\hat{A}^\pi(s_t, a_t) - bD_{KL}(\pi_\theta(\cdot|s_t) \parallel \pi(\cdot|s_t))\right] \tag{12}$$

which is also proposed in Schulman et al. (2017). In Figure 3 (b), we can observe that optimizing equation 12 still fails to prevent the probability ratio growing exponentially. While probability ratio clipping has proven insufficient to overcome numerical instability, it is worth noting that the PPO algorithm becomes even more unstable without clipping, as shown in Figure 3 (c) where the gradient explodes within 4 steps. When plotting $\max_t \log_{10}\left(p_\theta(a_t|s_t)\right)$, it is calculated after clipping. For example, a large $p_\theta(a_t|s_t)$ with positive advantage $\hat{A}^\pi(s_t, a_t)$ is clipped and equal to $1 + \epsilon$.

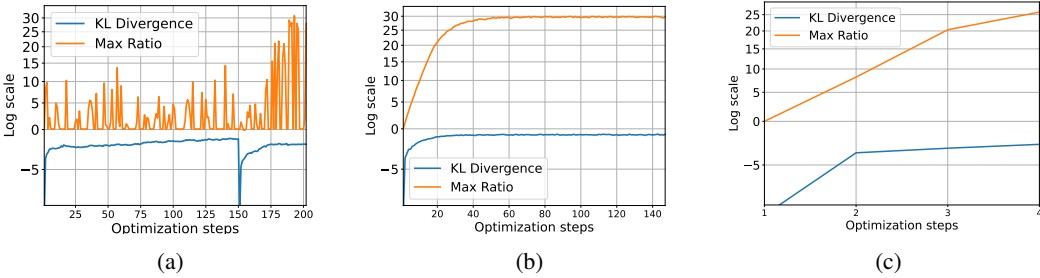

(a)                    (b)                    (c)

Figure 3: The logarithm of KL divergence $\log_{10}\left(D_{KL}(\pi_\theta(\cdot|s) \parallel \pi(\cdot|s))\right)$ and the maximum of probability ratio $\max_t \log_{10}\left(p_\theta(a_t|s_t)\right)$ at each optimization step is plotted with different optimization constraints, where environment is MuJoCo Humanoid-v4. In each plot, we apply (a) probability ratio clipping; (b) KL-penalty; (c) no constraints.

## 6 BYPASSING IMPORTANCE SAMPLING IN PPO

In previous sections, we demonstrated that the TRPO/PPO objective is potentially ill-conditioned and that the built-in optimization constraints cannot effectively alleviate its numerical instability. In this section, we examine the numerical stability of the vanilla policy gradient algorithm and propose a new algorithm based on it.

**Numerical stability of vanilla policy gradient.** We have discussed why importance sampling in TRPO/PPO is the primary cause of gradient explosion. It is also worth stepping back to revisit the vanilla policy gradient, whose objective does not involve any density ratio. In this case, we optimize the following objective

$$L^{VANILLA}(\theta) = \hat{\mathbb{E}}_{(s_t, a_t) \sim \pi_\theta}\left[\log \pi_\theta(a_t|s_t)\,\hat{A}^\pi(s_t, a_t)\right], \tag{13}$$

where we replace $Q^\pi(s_t, a_t)$ with $\hat{A}^\pi(s_t, a_t)$ to reduce the variance. The logarithm of the Gaussian probability density function is

| ENVIRONMENT | FINAL RETURN | EXPLOSION RATE |
|---|---|---|
| HOPPER-V4 | $536 \pm 254$ | 0% |
| WALKER2D-V4 | $195 \pm 135$ | 0% |
| HUMANOID-V4 | $355 \pm 191$ | 0% |

Table 2: Empirical results of vanilla policy gradient in three MuJoCo environments. Despite its poor performance, vanilla policy gradient does not suffer from the exploding gradient issue. Hyperparameters are specified in Appendix A.

$$\log \pi_\theta(a_t|s_t) = -\frac{1}{2} \sum_{i=1}^{m} \frac{(a_t - \mu)_i^2}{\sigma_i^2} - \frac{m}{2} \log(2\pi) - \sum_{i=1}^{m} \log \sigma_i, \tag{14}$$

which is less likely to take extreme values where $(a_t - \mu)_i$ denotes the $i$-th element in $(a_t - \mu)$ and $\Sigma = \text{diag}(\sigma_1^2, ..., \sigma_m^2)$. While the only problematic thing is that $\Sigma$ might have small eigenvalues, some DRL libraries employ a lower bound to the standard deviation of the policy to avoid singularity issues as shown in Table 5. Furthermore, the numerical stability of vanilla policy gradient is not very sensitive to the dimension of the action space: in equation 14, the magnitude of $|\log \pi_\theta(a_t|s_t)|$ is roughly linear to the dimension $m$. Similarly, the magnitude of the integrand $\mathcal{L}$ in equation 17 is also approximately linear to the dimension of $\mathcal{A}$, implying that the probability ratio $p_\theta$ is approximately exponential to the dimension of $\mathcal{A}$ and thus much easier to explode as $m$ increases.

In Table 2, we observe that the vanilla policy gradient does not exhibit any numerical instability when the standard deviations are lower-bounded, further supporting the claim that importance sampling is the primary cause of gradient explosion in deep policy gradient methods. It should be noted that while extremely small standard deviations can still affect numerical stability, this issue can be effectively mitigated by setting a lower bound on the standard deviations, as is done in PPO (though it is not entirely effective).

**Vanilla Policy Gradient with Clipping.** Based on the previous analysis, the vanilla policy gradient algorithm is more stable than TRPO/PPO. Motivated by this observation, we propose an objective that combines the strengths of both approaches. This modification results in the following objective:

$$L^{CPG}(\theta) = \hat{\mathbb{E}}_{(s_t, a_t) \sim \pi} \left[ \min \left( \log p_\theta \, \hat{A}_t^\pi, \text{clip}(\log p_\theta, \log(1-\epsilon), \log(1+\epsilon)) \hat{A}_t^\pi \right) \right] \tag{15}$$

where $\log p_\theta = \log \pi_\theta - \log \pi$. Notably, the new objective can also be interpreted as a logarithmic variant of the PPO loss in equation 11. As shown in Figure 5 and Table 2, this method achieves performance comparable to PPO and outperforms the vanilla policy gradient. Also, when there is no exploding gradient issue, our algorithm can achieve similar performance to PPO. To examine the performance in the general case, we use smaller policy networks with width 64 which are less affected by numerical issues based on our analysis in Section 3, and run both algorithms in three benchmarks. As shown in Figure 4, both achieve similar performance across all three environments, indicating that our algorithm serves as a numerically robust alternative to PPO.

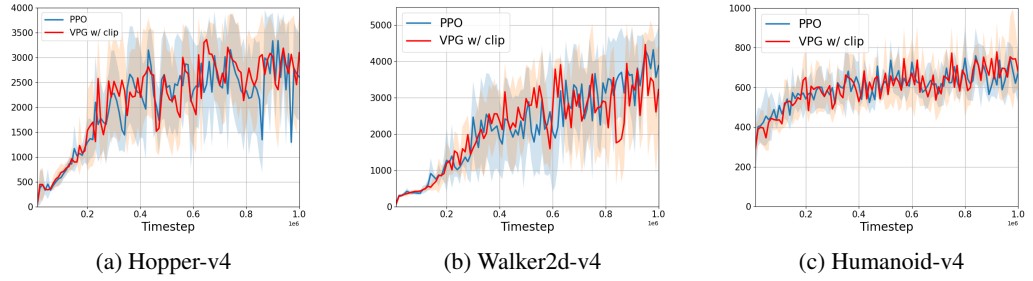

(a) Hopper-v4         (b) Walker2d-v4         (c) Humanoid-v4

Figure 4: For smaller policy networks that are less affected by gradient explosion, the proposed algorithm achieves performance comparable to PPO on three MuJoCo benchmarks.

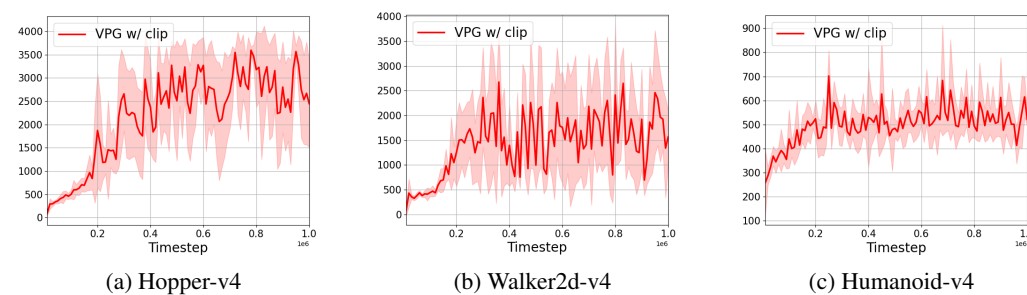

(a) Hopper-v4          (b) Walker2d-v4          (c) Humanoid-v4

Figure 5: For larger policy networks, where PPO completely fails due to gradient explosion, our algorithm avoids numerical issues and continues to produce reasonable results.

# 7 CONCLUDING REMARKS

In this work, we demonstrate that ill-conditioned density ratios in importance sampling are the fundamental cause of numerical instability in deep policy gradient methods. While this issue cannot be effectively addressed through optimization constraints alone, code-level techniques are necessary to fully eliminate or mitigate its effects. To address this issue, we find that directly dropping the importance sampling term yields the best performance, and we therefore advocate for further investigation in this direction in future studies.

According to our analysis, larger neural networks make bigger updates in each optimization step, resulting in larger density ratios in importance sampling and thus increasing the risk of gradient explosion. Therefore, we question whether deep policy gradient methods are numerically stable and robust when optimizing complex policy networks. This is particularly important for the scalability of deep policy gradient methods. Although the experiments in this work focus on MuJoCo continuous-control environments, the limitations discussed are inherent to the algorithms and may apply to other scenarios as well. We believe that a better understanding of these issues can undoubtedly enhance the effectiveness of deep reinforcement learning in real-world applications.

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

## A    EXPERIMENTAL SETUP

### A.1    DEFAULT HYPERPARAMETERS

|                                  | Hopper-v4 | Walker2d-v4   | Humanoid-v4   |
| -------------------------------- | --------- | ------------- | ------------- |
| Horizon                          | 1000      | 1000          | 1000          |
| Discount factor ($\gamma$)       | 0.99      | 0.99          | 0.99          |
| Num. epochs                      | 10        | 10            | 10            |
| Minibatch size                   | 64        | 64            | 64            |
| GAE factor ($\lambda$)           | 0.95      | 0.95          | 0.95          |
| Optimizer                        | Adam      | Adam          | Adam          |
| Learning rate                    | $10^{-4}$ | $10^{-4}$     | $10^{-4}$     |
| Clipping parameter $\epsilon$    | 0.2       | 0.2           | 0.2           |
| KL penalty coefficient $b$       | 0.01      | 0.01          | 0.01          |
| Advantage normalization          | False     | False         | False         |
| Policy network                   | [1024]    | [1024, 1024]  | [1024, 1024]  |
| Value network                    | [64, 64]  | [64, 64]      | [64, 64]      |
| Activation function              | tanh      | tanh          | tanh          |
| Gradient clipping ($l_2$ norm)   | 1.0       | 1.0           | 1.0           |

Table 3: Default PPO hyperparameters for all environments.

|                                  | Hopper-v4 | Walker2d-v4   | Humanoid-v4   |
| -------------------------------- | --------- | ------------- | ------------- |
| Horizon                          | 1000      | 1000          | 1000          |
| Discount factor ($\gamma$)       | 0.99      | 0.99          | 0.99          |
| GAE factor ($\lambda$)           | 0.95      | 0.95          | 0.95          |
| Optimizer                        | Adam      | Adam          | Adam          |
| Learning rate                    | $10^{-4}$ | $10^{-4}$     | $10^{-4}$     |
| Advantage normalization          | False     | False         | False         |
| Policy network                   | [1024]    | [1024, 1024]  | [1024, 1024]  |
| Value network                    | [64, 64]  | [64, 64]      | [64, 64]      |
| Activation function              | tanh      | tanh          | tanh          |
| Gradient clipping ($l_2$ norm)   | 1.0       | 1.0           | 1.0           |

Table 4: Default vanilla policy gradient hyperparameters for all environments.

| LIBRARY            | LOWER BOUND |
| ------------------ | ----------- |
| STABLE BASELINES3  | NOT APPLIED |
| RLLAB              | $10^{-6}$   |
| TORCHRL            | $10^{-4}$   |

Table 5: Lower bounds for the standard deviation in different DRL libraries.

### A.2    EXPERIMENTAL DESIGN IN SECTION 5

The initial policy $\pi(\cdot|s) = \mathcal{N}(\mu(s); \Sigma)$ where the covariance matrix $\Sigma = \mathcal{I}_{17 \times 17}$ is the identity matrix, its mean $\mu(s)$ is generated by

$$\mu(s) = f(s; \theta) + c$$

where $f$ is a policy network and $c = 100$ is a large constant that shifts $\mu(s)$ out of the action space $\mathcal{A}$ and makes the probability ratio easier to explode for the ease of illustration.

### A.3 STANDARD DEVIATION CLIPPING

To prevent the variance of the Gaussian policy $\pi$ from getting too close to $0$, the covariance matrix is parameterized as

$$\Sigma = \text{diag}(l(\sigma_1)^2, ..., l(\sigma_m)^2),$$

where $\sigma_i \in \mathcal{R}$ are directly optimized through PPO and $l(\sigma_i) = \text{softplus}(\sigma_i) + c_0$ where $c_0 = 0.1$.

## B ADDITIONAL EXPERIMENTS

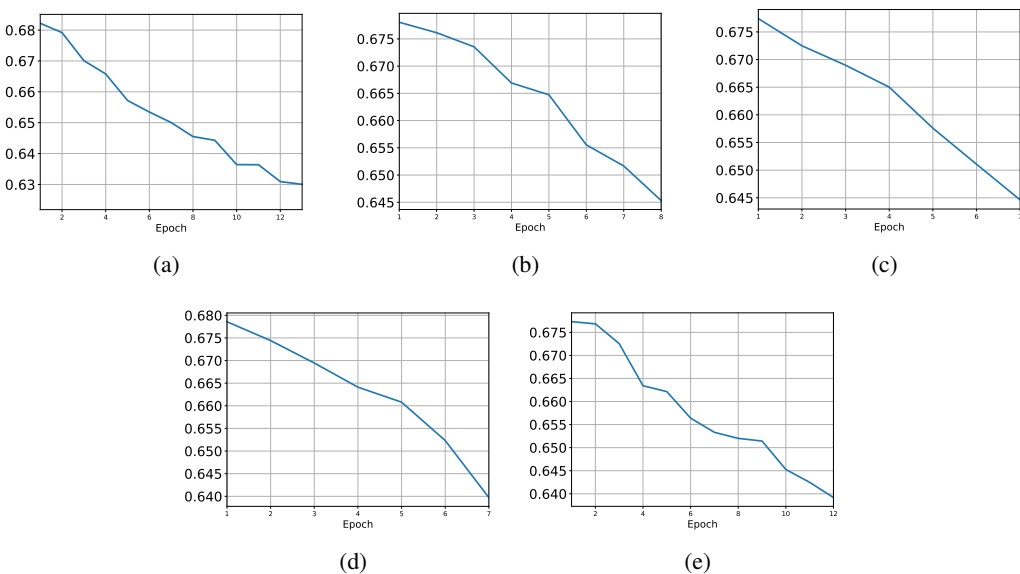

Figure 6: The corresponding minimum standard deviation of the Gaussian policy in the experiments shown in Figure 2. Standard deviations $\sigma = [\sigma_1, ..., \sigma_m]$ are state-independent and the minimum standard deviation is $\sigma_{min} = \min(\sigma_1, ..., \sigma_m)$.

**Reward scaling.** We will demonstrate that having high rewards is not the primary cause of gradient explosion in DRL, in contrast to what is suggested in GAIL, as noted in Wang et al. (2024). The results in Table 6 show that the explosion rate does not monotonically decrease with the scaling factor. Specifically, the explosion rate for a scaling factor of $0.001$ is even higher than that for a factor of $0.1$, suggesting that large reward values may not be the primary cause of exploding gradients, although they may have a significant impact on the final return. The reason is that arithmetic overflows are typically caused by exponentially large quantities, and most reward functions are not large enough to trigger gradient explosion.

| SCALING FACTOR | FINAL RETURN | MAXIMAL RETURN | EXPLOSION RATE |
| --- | --- | --- | --- |
| $10^0$ | N/A $\pm$ N/A | $1057 \pm 593$ | $100\%$ |
| $10^{-1}$ | $2792 \pm 792$ | $2897 \pm 759$ | $20\%$ |
| $10^{-2}$ | $2366 \pm 931$ | $3167 \pm 421$ | $40\%$ |
| $10^{-3}$ | $2303 \pm 0$ | $2269 \pm 933$ | $80\%$ |

Table 6: Final return, maximal return and explosion rate are reported for each scaling factor in the Hopper-v4 environment. 'N/A' indicates that the algorithm fails to complete training due to gradient explosion in all 5 individual runs. The final return is calculated only for successful runs, while the maximum return is calculated for every trial, whether completed or not. The gradient is considered exploded when the algorithm returns 'NaN' and/or 'Inf'.

**Learning rate and optimizer.** According to equation 7, the growth in $p_\theta(a|s)$ also depends on the total distance traveled along the curve $\mathcal{C}$.

$$p_{\theta_1}(\phi(a)|s) = \exp\Big(\int_{\mathcal{C}} (\phi(a) - \mu(s;\theta))^T \Sigma(\theta)^{-1} \frac{\partial \mu(s;\theta)}{\partial \theta} \cdot d\mathbf{r}\Big)$$

$$\simeq \exp\Big(\sum_{i=0}^{H} (\phi(a) - \mu(s;\theta_i))^T \Sigma(\theta_i)^{-1} \frac{\partial \mu(s;\theta_i)}{\partial \theta} \cdot \alpha\Big)$$

where $\alpha$ is the learning rate. It is clear that if we fix the number of epochs and training steps per epoch (i.e., $H$ is fixed), using a smaller $\alpha$ can reduce the likelihood of excessively large $p_\theta$ and exploding gradients. Nevertheless, it's important to recall that a fundamental distinction between PPO and the vanilla policy gradient method is that *the sampling policy is updated after a full epoch, rather than immediately after every optimization step*, which may contribute to performance improvements. Moreover, it has been proven in Schulman et al. (2015a) that

$$L^{VANILLA}(\theta_0) = L(\theta_0), \quad \nabla L^{VANILLA}(\theta_0) = \nabla L(\theta_0),$$

This implies that PPO converges to the vanilla policy gradient as $\alpha \to 0$, with the number of epochs and optimization steps kept constant,In Figure 7 (a), we observe that the performance of PPO deteriorates as the learning rate decreases. Another issue is the use of the Adam optimizer, which can cause large updates in the function space even with a small learning rate, particularly when the policy network is large. To illustrate this, consider the first timestep in Adam (Kingma & Ba, 2015):

$$\theta_1 = \theta_0 + \alpha \frac{\nabla L(\theta)}{\sqrt{\nabla L(\theta)^2 + \epsilon_0}}, \tag{16}$$

where all operations on vectors are element-wise and $\epsilon_0 \ll 1$ is a small positive quantity, which is approximately equivalent to

$$\theta_1 \simeq \theta_0 + \alpha \, \text{sgn}(\nabla L(\theta)),$$

meaning it does not directly bound the Euclidean distance between $\theta_0$ and $\theta_1$. The following example shows why the choice of optimizer can affect the size of the updating steps:

**Example B.1.** *Let $\eta = [\eta_1, ..., \eta_N] \in \mathbb{R}^N$ and $D = [-1, 1]$, $f_1, ..., f_N \in L^2(D)$ are $N$ functions that satisfy: (i) $\|f_i\|_{L^2(D)} = 1$ for all $i = 1, 2, ..., N$; (ii) $\langle f_i, f_j \rangle_{L^2(D)} = 0$ for all $i, j = 1, 2, ..., N$ and $i \neq j$. Define the parameterization mapping $\Phi : \mathbb{R}^N \to L^2(D)$ such that $\Phi(\eta) = \sum_{i=1}^{N} \eta_i f_i$. Let $\eta' = [\eta'_1, ..., \eta'_N] \in \mathbb{R}^N$ be another parameter and we have*

- *if $\|\eta' - \eta\|_2 = \alpha$, the $L^2$-distance in the function space $\|\Phi(\eta') - \Phi(\eta)\|_{L^2(D)} = \sqrt{\sum_{i=1}^{N} |\eta_i - \eta'_i|^2} = \alpha$;*

- *if $|\eta'_i - \eta_i| = \alpha$ for all $i$, the $L^2$-distance in the function space $\|\Phi(\eta') - \Phi(\eta)\|_{L^2(D)} = \sqrt{\sum_{i=1}^{N} |\eta_i - \eta'_i|^2} = \sqrt{N}\alpha$;*

*where $N$ is the dimension of the parameter space.*

Therefore, the update in the function space made by Adam heavily depends on the dimension of the policy space. In Figure 7 (b) and (c), we observe that as the network width increases, the growth rate of the $L^2$-distance between the current and initial network under the Adam optimizer grows faster than with SGD. This implies that the update from $\pi$ to $\pi_\theta$ may not be as small as suggested by the learning rate $\alpha$, potentially leading to excessively large values in $p_\theta$.

## C  IMPORTANCE SAMPLING IN RL

Here we study the dynamics of the probability ratio $p_\theta$ in the general case of parameterized standard deviations.

Assume that the covariance matrix $\Sigma(\theta) = \text{diag}(\sigma_1^2, ..., \sigma_m^2)$ and $\Sigma(\theta_0) = \text{diag}(\sigma_{0,1}^2, ..., \sigma_{0,m}^2)$. To avoid ambiguity, we further write $\theta = [\eta, \sigma_1, ..., \sigma_m]$ where $\eta$ denotes the policy parameterization in

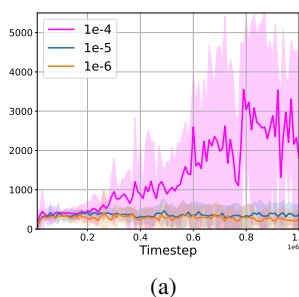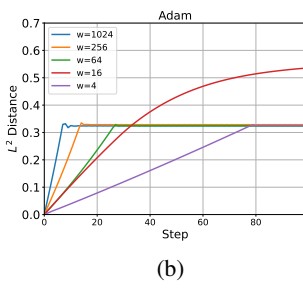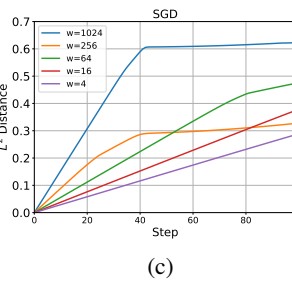

|     |     |     |
| --- | --- | --- |
| (a) | (b) | (c) |

Figure 7: (a) Comparison of different learning rates on the Walker2d-v4 environment. (b) & (c) The $L^2$ distance between the current network $f(\cdot; \theta_k)$ and initial network $f(\cdot; \theta_0)$ at step $k$ is compared for different widths $w$ and optimizers.

the mean $\mu = \mu(s; \eta)$. The probability ratio is given by

$$
p_\theta(\phi(a)|s) = \frac{\pi_\theta(\phi(a)|s)}{\pi(\phi(a)|s)}
$$

$$
= \sqrt{\frac{\det \Sigma(\theta_0)}{\det \Sigma(\theta)}} \frac{\exp\left(-\frac{1}{2}(\phi(a) - \mu(s;\eta))^T \Sigma(\theta)^{-1}(\phi(a) - \mu(s;\eta))\right)}{\exp\left(-\frac{1}{2}(\phi(a) - \mu(s;\eta_0))^T \Sigma(\theta_0)^{-1}(\phi(a) - \mu(s;\eta_0)))\right)}.
$$

$$
= \frac{\sigma_{0,1}...\sigma_{0,m}}{\sigma_1...\sigma_m} \exp\left(\frac{1}{2}\sum_{i=1}^m \sigma_{0,i}^{-1}(\phi(a)_i - \mu(s;\eta_0)_i)^2 - \frac{1}{2}\sum_{i=1}^m \sigma_i^{-1}(\phi(a)_i - \mu(s;\eta)_i)^2\right)
$$

where we use subscriptions to denote the corresponding element in a vector. For instance, $\phi(a) = [\phi(a)_1, ..., \phi(a)_m]$ has $m$ elements and $\phi(a)_i$ denotes the $i$-th term.

Similar to the constant variance case in the main text, the gradient of $p_\theta$ with respect to the mean parameterization is

$$
\frac{\partial}{\partial \eta} \log p_\theta(\phi(a)|s) = (\phi(a) - \mu(s;\eta))^T \Sigma(\theta)^{-1} \frac{\partial \mu(s;\eta)}{\partial \theta},
$$

which has the same dynamics described in Section 3. The gradient of $p_\theta$ with respect to $\sigma_i$ is given by

$$
\frac{\partial}{\partial \sigma_i} p_\theta(\phi(a)|s) = -\frac{1}{\sigma_i} p_\theta(\phi(a)|s) + p_\theta(\phi(a)|s) \cdot \frac{1}{2\sigma_i^2}(\phi(a)_i - \mu(s;\eta)_i)^2
$$

$$
= p_\theta(\phi(a)|s) \frac{(\phi(a)_i - \mu(s;\eta)_i)^2 - 2\sigma_i}{\sigma_i^2}
$$

Dividing $p_\theta(\phi(a)|s)$ both sides yields

$$
\frac{\partial}{\partial \sigma_i} \log p_\theta(\phi(a)|s) = \frac{(\phi(a)_i - \mu(s;\eta)_i)^2 - 2\sigma_i}{\sigma_i^2}.
$$

Therefore, the full dynamics of $p_\theta$ along the training curve $\mathcal{C}$ is given by

$$
p_\theta(\phi(a)|s) = \exp\left(\int_{\mathcal{C}} \mathcal{L}(s, \phi(a); \theta) \cdot d\mathbf{r}\right), \tag{17}
$$

where $\mathcal{L}(s, \phi(a); \theta)$ is the full gradient whose elements are given $p_\theta$ is still exponential to some line integral in the general case, meaning that the discussions in the main text applies to common practice.

## D   THEORETICAL FOUNDATIONS OF POLICY GRADIENT METHODS

**Policy Gradient Theorem.** In its original formulation (Sutton et al., 1999), the theorem states that the gradient of objective function $L(\theta)$ with respect to the policy parameter can be estimated through:

$$
\nabla_\theta L(\theta) \propto \int_{\mathcal{S}} \rho^\pi(s) \int_{\mathcal{A}} Q^\pi(s, a) \nabla_\theta \pi_\theta(a|s) \, da \, ds, \tag{18}
$$

In the integral form above, $\rho^\pi(\cdot)$ is the discounted visitation density under $\pi$ and $Q^\pi$ is the $Q$-function of $\pi$. A same theorem was established for deterministic policies under similar smoothness assumptions on value function and policy parameterization (Silver et al., 2014; Lillicrap et al., 2015).

**Failure of smoothness assumption.** In the original proof of the policy gradient theorem, the gradient estimator equation 18 is shown exact using the fact that $\gamma^t \|\nabla_\theta V^\pi(s)\| \to 0$ uniformly for all $s \in \mathcal{S}$ as $t \to \infty$, which then leads to the vanishing tail term in the rollout. Actually, it implicitly assumes that

- *(Smoothness Assumption) $\nabla_\theta V^\pi(s)$ exists and is continuous over $\mathcal{S}$.*

This assumption is proved valid in the case of finite state-space and stochastic policies (Agarwal et al., 2021). However, we will show that it is not generally true.

**Maximal Lyapunov Exponents.** Consider the system

$$s_{t+1} = F(s_t), \quad s_0 \in \mathbb{R}^N,$$

and a small perturbation $\Delta Z_0$ to $s_0$. The resulted divergence under $\Delta Z_0$ at time $t$ is denoted by $\Delta Z(t)$. For chaotic systems, their dynamics are sensitive to initial conditions so that it has

$$\|\Delta Z(t)\| \simeq e^{\lambda t} \|\Delta Z_0\|$$

for some positive $\lambda$ that is called the Lyapunov exponent Lorenz (1995). To make it precise, we present the definition of maximal Lyapunov exponents (MLEs):

**Definition D.1.** *(Maximal Lyapunov exponent) For the dynamical system $s_{t+1} = F(s_t), s_0 \in \mathbb{R}^n$, the maximal Lyapunov exponent $\lambda_{\max}$ at $s_0$ is defined as the largest value such that*

$$\lambda_{\max} = \limsup_{t \to \infty} \limsup_{\|\Delta Z_0\| \to 0} \frac{1}{t} \log \frac{\|\Delta Z(t)\|}{\|\Delta Z_0\|}. \tag{19}$$

The policy gradient theorem has been shown no longer true when the underlying dynamics is chaotic as in many continuous-control environments:

**Theorem D.1.** *(Fractal Landscapes in RL, Wang et al. (2023)) Assume that the dynamics, reward function and policy are all Lipschitz continuous with respect to their input variables. Let $\pi_\theta$ be a deterministic policy and $\lambda(\theta)$ denote the maximal Lyapunov exponent of the dynamics. Suppose that $\lambda(\theta) > -\log\gamma$, then*

1. *$V^{\pi_\theta}(s)$ is $\frac{-\log\gamma}{\lambda(\theta)}$-Hölder continuous in the state $s \in \mathcal{S}$;*

2. *$Q^{\pi_\theta}(s,a)$ is $\frac{-\log\gamma}{\lambda(\theta)}$-Hölder continuous in the action $a \in \mathcal{A}$;*

3. *$L(\theta)$ is $\frac{-\log\gamma}{\lambda(\theta)}$-Hölder continuous in the policy parameter $\theta \in \mathbb{R}^N$.*

Specifically, we say that a mapping $f : \mathbb{R}^N \to \mathbb{R}^m$ is $\alpha$-Hölder continuous at $x = x_0$ if there exists $K, \delta > 0$ such that $|f(x) - f(x_0)| \le K|x - x_0|^\alpha$ for all $x \in \mathbb{R}^N$ with $|x - x_0| \le \delta$. It reduces to Lipschitz continuity when $\alpha = 1$.

# E  LOSS FUNCTIONS IN SUPERVISED LEARNING

Here we briefly discuss why supervised learning usually does not suffer from exploding gradient issues. Consider a simple case where $\mathcal{X} = \{x_1, ..., x_M\} \subset \mathbb{R}^p$ and $\mathcal{Y} = \{y_1, ..., y_M\} \in \mathbb{R}^q$ are the data and label sets, respectively. The mean-squared error (MSE) is given as

$$L(\theta) = \frac{1}{M} \sum_{i=1}^{M} |f(x_i; \theta) - y_i|^2, \tag{20}$$

where $f(\cdot;\theta)$ is a neural network to fit and $\theta$ denotes its parameters. Suppose that all data points are uniformly sampled from a compact set $D \subset \mathbb{R}^p$, when the sample size is sufficiently large, we have

$$L(\theta) = \frac{1}{M}\sum_{i=1}^{M}|f(x_i;\theta) - y_i|^2 \simeq \int_D |f(x;\theta) - \phi(x)|^2 \, \mathrm{d}x$$

where $\phi : \mathbb{R}^p \to \mathbb{R}^q$ is the target mapping that generates the labels (assume it exists). Therefore, the gradient of $L(\theta)$ converges to

$$\nabla L(\theta) \simeq \frac{\partial}{\partial\theta}\int_D |f(x;\theta) - \phi(x)|^2 \, \mathrm{d}x.$$

Using the fact that a neural network $f(x;\theta)$ satisfies

- For almost every $x$, $\frac{\partial f(x;\theta)}{\partial\theta}$ exists for all $\theta$;

- $\|\frac{\partial f(x;\theta)}{\partial\theta}\|$ is bounded on any compact sets.

Note that we *do not need any smoothness assumptions on $\phi$ beyond integrability*. According to the Leibniz integral rule, it allows to switch the integral and differentiation, i.e.,

$$\nabla L(\theta) \simeq \frac{\partial}{\partial\theta}\int_D |f(x;\theta) - \phi(x)|^2 \, \mathrm{d}x = \int_D \frac{\partial}{\partial\theta}|f(x;\theta) - \phi(x)|^2 \, \mathrm{d}x,$$

which guarantees the convergence of the objective gradient $\nabla L(\theta)$ as the sample size $M \to \infty$. Therefore, unlike reinforcement learning whose objective function may have fractal landscapes in many robotics environments, supervised learning always has a differentiable objective which allows gradient-based algorithms to optimize.

**Inexact value and advantage estimations.** It should be noted that the numerical instability caused by the probability ratio can be avoided if both the value and advantage functions are estimated precisely. Suppose that the value function $V^\pi(\cdot)$ and the advantage estimate $A^\pi(\cdot,\cdot)$ are exact for the old policy $\pi$. For a given state $s$, assume that the mean $\mu(s)$ of $\pi$ is sufficiently large so that nearly all actions sampled from $\pi(\cdot|s)$ are clipped to the same action $\bar{a} = \phi(a)$. Note that the advantage

$$A^\pi(s,\bar{a}) = r(s,\bar{a}) + \gamma\mathbb{E}_{(s,\bar{a})\to s'}[V^\pi(s')] - \mathbb{E}_{a\sim\pi}[r(s,\phi(a)) + \gamma V^\pi(s'')]$$
$$= r(s,\bar{a}) - \mathbb{E}_{a\sim\pi}[r(s,\phi(a))] + \gamma(\mathbb{E}_{(s,\bar{a})\to s'}[V^\pi(s')] - \mathbb{E}_{a\sim\pi,(s,\phi(a))\to s''}[V^\pi(s'')]),$$

where $P(\phi(a) = \bar{a} \,|a \sim \pi) \simeq 1$ due to the large mean and action-clipping transformation, which further implies $\mathbb{E}_{a\sim\pi}[r(s,\phi(a))] \simeq r(s,\bar{a})$ and $\mathbb{E}_{a\sim\pi,(s,\phi(a))\to s''}[V^\pi(s'')] \simeq \mathbb{E}_{(s,\bar{a})\to s'}[V^\pi(s')]$. Therefore, it yields

$$A^\pi(s,\bar{a}) \simeq r(s,\bar{a}) - r(s,\bar{a}) + \gamma(\mathbb{E}_{(s,\bar{a})\to s'}[V^\pi(s')] - \mathbb{E}_{(s,\bar{a})\to s'}[V^\pi(s')]) = 0$$

which means that the advantage $A^\pi(s,\bar{a})$ at state $s$ should be very close to $0$ when the old policy already has a large mean $\mu(s)$. However, in practice, the error in advantage estimation can be significant. Value approximation may also be poor, as the true value landscape in many continuous control environments is often highly non-smooth and even fractal (Figure 8 (a)), whereas the value function estimated by neural networks is usually smooth (Figure 8 (b)). To test the accuracy of advantage estimation, we adopt the experimental setting from Section 5 and initialize the policy network with a large positive constant added to its output, ensuring that the mean stays far from the action space. In Figure 8 (c), it can be observed that the mean of the absolute values of the advantage estimated by GAE, i.e., $\frac{1}{T}\sum_{t=0}^{T-1}|\hat{A}^\pi(s_t, a_t)|$, remains around 1, even when the maximum of the probability ratio, $\max_t \log_{10}\left(p_\theta(a_t|s_t)\right)$, becomes very large.

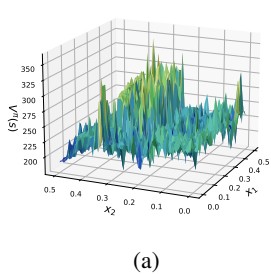 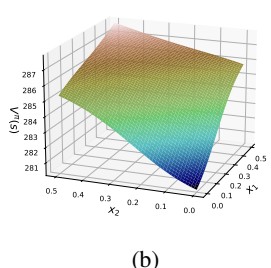 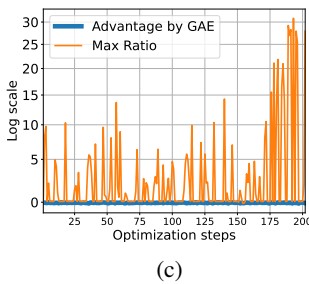

(a)                                (b)                                (c)

Figure 8: (a) An evident fractal structure in the true value landscape can be observed in the Hopper-v4 environment, resulting from the chaotic nature of the underlying dynamics. (b) The value function approximated by a neural network, however, is smooth. (c) The advantage estimation obtained from GAE does not decay to zero, even when the mean $\mu(s)$ is far from the action space $\mathcal{A}$, and is therefore unable to counterbalance the large values in the probability ratio. All values in Figure (c) are presented on the $\log_{10}$ scale.

## F    PROOFS

### F.1    DERIVATION OF EQUATION 4

Let $\Sigma = \sigma^T \sigma$ be the decomposition and $y = \sigma(a - \mu_0(s))$, then we have $y \sim \mathcal{N}(\mathbf{0}, \mathcal{I}_{m \times m})$ as the standard $m$-dimensional Gaussian variable and

$$
\begin{aligned}
P(\pi(a|s) < e) &= P(\frac{1}{\sqrt{(2\pi)^m \det \Sigma_0}} \exp(-\frac{1}{2}|y|^2) < e) \\
&= P(|y|^2 > -2\log(e\sqrt{(2\pi)^m \det \Sigma_0})) \\
&= P(y_1^2 + ... + y_m^2 > -2\log(e\sqrt{(2\pi)^m \det \Sigma_0}))
\end{aligned}
$$

where $y = [y_1, ..., y_m]^T$ and $y_i \sim \mathcal{N}(0,1)$ for all $i = 1, 2, ..., m$. Let $C = -2\log(e\sqrt{(2\pi)^m \det \Sigma_0})$ and it further has

$$
\begin{aligned}
P(y_1^2 + ... + y_m^2 > C) &\leq \sum_{i=1}^{m} P(y_i^2 > \frac{C}{m}) \\
&= mP(y_1^2 > \frac{C}{m}) \\
&= 2mP(y_1 > \sqrt{\frac{C}{m}}).
\end{aligned}
$$

Applying the tail bound $P(y_1 > t) \leq \exp(-\frac{t^2}{2})$ for standard Gaussian distribution yields

$$
\begin{aligned}
P(y_1 > \sqrt{\frac{C}{m}}) &\leq \exp(-\frac{C}{2m}) \\
&= \sqrt{2\pi}(e\sqrt{\det \Sigma_0})^{\frac{1}{m}}.
\end{aligned}
$$

Substituting this into the previous equality yields

$$
P(\pi(a|s) < e) \leq 2\sqrt{2\pi}m(e\sqrt{\det \Sigma_0})^{\frac{1}{m}},
$$

and we complete the proof.

### F.2    PROOF OF THEOREM 3.1

*(I)* $\mu(s) \in \mathcal{A}$: Note that the distance between clipped action $\phi(a)$ and the mean $\mu(s)$ is no more than the original distance, i.e., $|\phi(a) - \mu(s)| \leq |a - s|$. Applying equation 4 immediately yields the result.

*(II)* $\mu(s) \notin \mathcal{A}$: We have

$$\pi(\phi(a)|s) = \frac{1}{(2\pi)^{\frac{m}{2}}\sqrt{\det \Sigma}} \exp\left(-\frac{(\mu(s) - \phi(a))^T \Sigma^{-1}(\mu(s) - \phi(a))}{2}\right)$$

$$\leq \frac{1}{(2\pi)^{\frac{m}{2}}\sqrt{\det \Sigma}} \exp\left(-\frac{|\mu(s) - \phi(a)|\|\Sigma^{-1}\|_2|\mu(s) - \phi(a)|}{2}\right)$$

$$= \frac{1}{(2\pi)^{\frac{m}{2}}\sqrt{\det \Sigma}} \exp\left(-\frac{\frac{1}{\lambda_{max}}|\mu(s) - \phi(a)|^2}{2}\right)$$

$$\leq \frac{1}{(2\pi)^{\frac{m}{2}}\sqrt{\det \Sigma}} \exp\left(-\frac{d^2}{2\lambda_{max}}\right)$$

using the fact that $\|\Sigma^{-1}\|_2 = \|\Sigma\|_2^{-1} = \frac{1}{\lambda_{max}}$ and $|\mu(s) - \phi(a)| \geq \min_{y \in \mathcal{A}} |\mu(s) - y| = d$.

