# OpenReview forum: "Numerical Pitfalls in Policy Gradient Updates"
_ICLR.cc/2025/Conference — Submitted to ICLR 2025_

### Official Review · Reviewer_uwTa · 2024-10-29

**Soundness:** 2
**Presentation:** 2
**Contribution:** 2
**Rating:** 6
**Confidence:** 2

**Summary:**

This paper addresses the issue of numerical instability in deep reinforcement learning. It begins by positing that importance sampling weights are the primary cause of exploding gradients in PPO and TRPO. A series of example calculations is then used to demonstrate why the importance sampling factor can lead to exploding gradients. Finally, the paper presents potential solutions to ensure numerical stability, along with a discussion of their respective advantages and disadvantages.

**Strengths:**

Numerical instability is an important topic, and understanding its root cause will help improve the practical application of RL.

**Weaknesses:**

I found it challenging to identify a distinctly new contribution in this paper. The issue of numerical instability due to importance sampling is well recognized, and several of the potential solutions explored here have also been discussed in prior work. While the deeper mathematical analysis of these causes is undoubtedly helpful, I am not entirely convinced that the contribution provides enough novelty in its current form.

W1: The analysis seems to be limited to gaussian policies.

W2: From a statistical point of view, using just three environments to support your claims is insignificant. In my view, a case study with a statistically significant sample size would be necessary to substantiate your hypotheses effectively.

W3: I found it difficult to understand the figures from the captions, and at times, the text didn’t clarify either. For example, in Fig. 1(a), what exactly is being plotted? The condition number $\kappa(\theta)$ requires a function $f$ (which I assume is $p_\theta$ here), but what do $\mu_0$ and $\mu$ refer to, respectively? Please also see additional questions below.

**Questions:**

Q1: What changes when other policy parameterisations are used e.g. beta distribution reshaped to fit the desired action range. Then clipping becomes unnecessary. Is the importance sampling weight still the root course of exploding gradients?

Q2: I understand why large importance sampling rates cause numerical issues, but I didn’t immediately see the connection to exploding gradients, as a large objective value does not directly imply a large gradient. This question remained until Section 3, in the paragraph titled "Ill-conditioned probability ratio," where you explain the link between large importance sampling weights and large gradients (or a large condition number) for Gaussian policies. Could you elaborate on this in more detail for general parameterizations? This explanation is fundamental to the understanding of the paper and might be discussed in the beginning of Sec. 3.

Q3: What happens when a small epsilon is added in the denominator of the importance sampling weight? Specifically, if we have $p_\theta(a|s) = \frac{\pi_\theta(a|s)}{\pi(a|s) +\epsilon}$, this would ensure numerical stability for the objective function and would be the first approach I would consider to address the issue. However, what is the effect on the gradients?

Q4: In line 379, you state that a large mean always leads to the same clipped action. I don’t see how this is true in general; this only holds if the mean is far outside the action space, which is probably what you meant?

Q5: Fig. 3(b): What neural network is used to approximate the value function? If I am informed correctly, neural networks are only smooth when analytic activation and output functions are used. Especially using ReLU leads to non smooth functions, right?

Q6: Fig. 4.: What is plotted in (a); What is the y-axis? What causes the constant lines in (b)? Is the algorithm converged?


**Remarks on Notation:**

R1: line 431: The vanilla objective was not yet introduced.

R2: line 432: what is $\alpha$? Is it the step size as later in (12)?


-----------------------------------------------------------------------
Update: I would like to thank the authors for their thoughtful revision.

Upon reviewing the updated version of the paper, I noticed that the manuscript now conveys a completely new message. In my understanding, the primary contribution of the paper lies in the newly proposed policy optimization method presented in Equation (15). This method replaces the importance sampling ratio in PPO with the logarithm of the importance sampling weight.

The authors argue that this approach addresses numerical instability, which is a notable improvement. While I am not deeply familiar with every variation of policy gradient methods, I cannot confidently confirm whether this specific approach has been proposed before. However, if the method is indeed novel, I find it to be a highly interesting contribution that would be valuable to share with the community. I raised my score to 6.

---

> ### Author Response · Authors · 2024-11-18
> **Rebuttal**
>
> Thanks for the detailed feedback, we hope that the following response can address your concerns.
>
> For the weaknesses:
>
> > The issue of numerical instability due to importance sampling is well recognized, and several of the potential solutions explored here have also been discussed in prior work.
>
> We agree that the issue of numerical instability due to importance sampling has been studied in prior work, such as [1], as discussed in the Related Work section. Our work differs in that, while previous studies highlighted how the product of density ratios over a long horizon can grow exponentially, leading to exploding variance in off-policy estimation, we demonstrate that even a single density ratio can take excessively large values and cause numerical overflow.
>
> To the best of our knowledge, no existing work explicitly relates arithmetic overflow in deep policy gradients to the numerical instability of importance sampling. Currently, this numerical issue is generally viewed as a hyperparameter tuning problem, as suggested in the Stable-Baselines3 documentation (https://stable-baselines3.readthedocs.io/en/master/guide/checking_nan.html). Our work contributes to this line of research by directly identifying the ill-conditioned nature of importance sampling as the root cause of the problem.
>
> Please also refer to the global response, where we summarize the contributions of our work.
>
> W1: Our analysis focuses on Gaussian policies because they are the default in policy gradient methods and most theoretical frameworks and algorithms are designed for this case. However, the discussion on the ill-conditionedness of importance sampling is not limited to Gaussian policies and can be applied to various probability distributions.
>
> W2: We agree that using more environments could further strengthen the analysis. However, we believe that the three environments (Hopper, Walker, Humanoid) considered in the paper are sufficient to support the theoretical analysis, as they were adopted from two well-known prior works [2, 3].
>
> W3: In Figure 1(a), it shows how the condition number $\kappa(\theta)$ changes with respect to the mean of $\pi$ ($\mu_0$), and the difference between the mean of $\pi_\theta$ ($\mu$) and $\mu_0$. The condition number is related to the probability ratio $p_\theta$. These plots indicate that the condition number of the importance sampling ratio grows exponentially as $\pi_\theta$ deviates from $\pi$, potentially resulting in arithmetic overflow. We thank the reviewer for pointing it out and now we have fixed it.
>
> **References:**
>
> [1] "Breaking the curse of horizon: Infinite-horizon off-policy estimation" by Liu et al., NeurIPS, 2018;
>
> [2] "Implementation Matters in Deep Policy Gradients: A Case Study on PPO and TRPO" by  Engstrom et al., ICLR, 2019;
>
> [3] "A closer look at deep policy gradients" by Ilyas et al., ICLR, 2020;
>
> For the questions:
>
> Q1: As we mentioned, the importance sampling ratio can become ill-conditioned when the probability density function takes excessively small values. This can occur with beta distributions, which have zero or infinite density at the boundaries. Therefore, importance sampling remains the root cause of exploding gradients, as it may involve dividing by extremely small values.
>
> Q2: They are related as follows: the importance sampling ratio can grow exponentially, leading to arithmetic overflow. When this occurs, the objective value to be differentiated becomes NaN, which corrupts the entire computational graph of auto-differentiation, thereby causing exploding gradient issues. This is why techniques like gradient clipping cannot prevent the issue—because the overflow happens before the gradient computation.
>
> Q3: We agree that it would ensure the numerical stability for PPO. One potential problem is that when both $\pi_\theta$ and $\pi$ are very small, the probability ratio will always take very small values and hence affect the clipping law in PPO (which assumes that the ratio stays near 1). We proposed a method in the revision that replaces the importance sampling ratio with its logarithm, and it works out.
>
> Q4: Yes, that is why we said "assume that the mean $\mu(s)$ is SUFFICIENTLY large".
>
> Q5: The network architecture used for the value function is described in Appendix A. While using ReLU leads to non-smooth functions, they are at least piecewise linear. However, the true value function, as shown there, can have fractal structures and be much more non-smooth than ReLU functions.
>
> Q6: In Figure 4 (now in the appendix), the y-axis in (a) represents the cumulative reward. The constant lines in (b) are due to the convergence of the algorithm.
>
> We also thank the reviewer for pointing out the notation issues, which have been fixed in the revised draft.

---

> ### Author Response · Authors · 2024-11-28
>
> Dear Reviewer uwTa,
>
> We are delighted to hear that the revised draft has addressed your concerns. Thank you for raising the score and for supporting our work!

---

### Official Review · Reviewer_1MG6 · 2024-11-02

**Soundness:** 2
**Presentation:** 3
**Contribution:** 2
**Rating:** 5
**Confidence:** 3

**Summary:**

The authors conduct an empirical study of a specific issue related to the importance weights in policy optimization, which is that the probability ratio of the current policy to the data collection policy is not necessarily upper bounded and can be arbitrarily large with a non zero probability. The paper hypothesizes that this could lead to numerical instability and goes on to propose ways to prevent this from occurring especially for feasible bounded continuous action sets. However, they observe that while this is successful at bounding the maximum probability ratio, this actually leads to worse final performance on the returns and then go on to speculate that this implies that the success of deep policy gradient methods may be attributable in part to the numerical instability. This doesn't seem like a valid inference to me, since the maximum probability density ratio might not explain what is happening in the non-extreme cases that constitute the bulk of the samples being learned from.

**Strengths:**

- Authors conduct a focused experimental study of a specific numerical issue that might not otherwise be highlighted in literature.
- The paper puts forth bold hypotheses and presents a good didactic discussion of issues related to clipping the action and how the mean being outside a feasible set can unintentionally lead to a large density ratio.

**Weaknesses:**

- The authors claim that the success of policy gradient algorithms is due to numerical instability -- this seems like an odd thing to infer from their ablations.
- While the point made in L137-140 about sampling from $\pi$ notwithstanding, clipping them to a bounded range leading to an unintentionally small value for $pi$, this doesn't seem like a strong enough case to justify the broad claim in the title in the section, "importance sampling considered harmful".
- Most of the insights contained in the paper are either not entirely novel or they are hard to be convinced of (e.g. claims about importance sampling being harmful, or numerical instability benefitting deep pg algorithms), from the empirical observations. There isn't much theory to speak of, and the main theorem is just a simple statement about the standard gaussian normal density.

**Questions:**

- Figure 2 unclear what is the experiment ablation being studied across the three variants (a), (b) and (c).
- In Table 2, can you describe "explosion rate" a bit more and how this impacts the learning? Is this the fraction of samples for which the gradients are NaN and in which case the sample is effectively discarded from the mini batch that computes the model update? Or is it the fraction of minibatches where the update is not computed, i.e. effectively a full training step wasted.

---

> ### Author Response · Authors · 2024-11-18
> **Rebuttal**
>
> Thank you for the feedback and please see the following responses to your questions.
>
> For the weaknesses:
>
> > The authors claim that the success of policy gradient algorithms is due to numerical instability -- this seems like an odd thing to infer from their ablations.
>
> In the revised draft, we have adjusted the wording to better emphasize the point that the importance sampling term may not be necessary in policy gradient methods.
>
> > While the point made in L137-140 about sampling from $\pi$ notwithstanding, clipping them to a bounded range leading to an unintentionally small value for pi, this doesn't seem like a strong enough case to justify the broad claim in the title in the section, "importance sampling considered harmful".
>
> Thank you for pointing that out. We have rewritten the section title accordingly.
>
> > Most of the insights contained in the paper are either not entirely novel or they are hard to be convinced of (e.g. claims about importance sampling being harmful, or numerical instability benefitting deep pg algorithms), from the empirical observations. There isn't much theory to speak of, and the main theorem is just a simple statement about the standard gaussian normal density.
>
> We have added more empirical analysis to the revised draft and proposed an algorithm that combines the merits of vanilla policy gradient and PPO. This algorithm effectively overcomes the numerical issues while achieving performance comparable to PPO. We believe that the contributions of this work are now clearly presented. Controversial statements have also been removed.
>
> For the questions:
>
> > Figure 2 unclear what is the experiment ablation being studied across the three variants (a), (b) and (c).
>
> In Figure 2 (now Figure 3), we empirically demonstrate that optimization constraints, such as clipping and KL penalties, are not sufficient to prevent the importance sampling ratio from exploding. The difference between these plots lies in the constraints applied to the policy training: (a) probability ratio clipping, (b) KL-penalty, and (c) no constraints. We have clarified this point in the revised draft.
>
> > In Table 2, can you describe "explosion rate" a bit more and how this impacts the learning? Is this the fraction of samples for which the gradients are NaN and in which case the sample is effectively discarded from the mini batch that computes the model update? Or is it the fraction of minibatches where the update is not computed, i.e. effectively a full training step wasted.
>
> In Table 2, the explosion rate represents the percentage of individual runs that encounter arithmetic overflow and return NaN. For example, if 2 out of 5 random seeds encounter gradient explosion, the explosion rate would be 40%.
>
> We are happy to answer any follow-up questions.

---

> > ### Author Response · Authors · 2024-11-28
> >
> > Dear Reviewer 1MG6,
> >
> > Thank you once again for your time and effort in reviewing our work! We hope our response has addressed your concerns. As the discussion period comes to a close, please let us know if you have any further questions.

---

> > > ### Comment · Reviewer_1MG6 · 2024-12-02
> > > **Reply**
> > >
> > > Thanks for the detailed replies. I will edit my score appropriately in short order.

---

> > > > ### Author Response · Authors · 2024-12-02
> > > >
> > > > Dear Reviewer 1MG6,
> > > >
> > > > Thank you for updating your score. If you have any further concerns, please do not hesitate to let us know so we can address them before the discussion period ends.

---

### Official Review · Reviewer_ikDh · 2024-11-03

**Soundness:** 3
**Presentation:** 2
**Contribution:** 2
**Rating:** 6
**Confidence:** 3

**Summary:**

Gradient explosion, during policy optimization is a significant problem in deep reinforcement learning (DRL). The authors present an analysis of why this issue occurs and propose directions for improving stability in policy gradient methods. The authors hypothesize that the root cause lies in the importance sampling step, specifically the density ratio in the surrogate objective function. They show that the TRPO/PPO loss is inherently ill-conditioned due to importance sampling and action clipping, which directly contribute to the numerical instability.

**Strengths:**

The paper is organized, progressing logically from problem definition to analysis, experiments, and suggested approaches.  By comparing PPO with TRPO and vanilla policy gradient, the authors effectively demonstrate that vanilla policy gradient methods, despite their simplicity, do not suffer from the same instability. This comparative approach highlights the specific issues with importance sampling and policy constraints.

**Weaknesses:**

FINAL RETURN is not a fully informative comparison measure. The hyperparameter tuning process in your experiments is not clearly explained, and five runs seem insufficient for reliable results. Increasing this to ten runs would provide more robust data. Additionally, were these runs conducted with different random seeds?

Another concern I have is regarding the content. I would have given a strong recommendation if the paper had introduced a rigorous algorithm to bypass the need for importance sampling in TRPO/PPO. Bypassing IS has been addressed in works such as 'Efficiently Escaping Saddle Points for Non-Convex Policy Optimization' (https://arxiv.org/pdf/2311.08914), which should have been referenced to provide context and highlight potential alternative approaches.

**Questions:**

Why did you use FINAL RETURN to report? why the final is important? not previous ones?

---

> ### Author Response · Authors · 2024-11-18
> **Rebuttal**
>
> Thanks for the positive review. Please see the following responses to your concerns.
>
> For the weaknesses:
>
> > FINAL RETURN is not a fully informative comparison measure. The hyperparameter tuning process in your experiments is not clearly explained, and five runs seem insufficient for reliable results. Increasing this to ten runs would provide more robust data. Additionally, were these runs conducted with different random seeds?
>
> We agree that using more random seeds would yield more robust data. Our experimental setup is primarily based on [1], which used five random seeds for each environment. Similarly, [2] also employed five seeds, balancing the computation budget with the need for robustness.
>
> Regarding the performance metric, we reported both FINAL RETURN and MAXIMAL RETURN when they differ significantly, and only FINAL RETURN when it is equal to (or nearly equal to) the MAXIMAL RETURN.
>
> For the hyperparameter tuning process, we followed the procedure outlined in Appendix A, which was adopted from previous works.
>
> In all experiments presented in Section 5 and 6, we fixed the random seeds (42–46) to ensure consistency and comparability of the results. This approach was also applied to the additional experiments detailed in Appendix B (now in Section 4).
>
> > I would have given a strong recommendation if the paper had introduced a rigorous algorithm to bypass the need for importance sampling in TRPO/PPO.
>
> That’s a great point! We just proposed a new solution called "vanilla policy gradient with clipping," which removes the importance sampling ratio from the PPO objective and applies the same clipping rule to the vanilla policy gradient objective. This algorithm combines the effectiveness of PPO with the robustness of VPG. The paper you suggested is highly relevant to the topic and we have discussed it in Related Work. Please refer to the global response and the revised draft for further details.
>
> For the question:
>
> > Why did you use FINAL RETURN to report? why the final is important? not previous ones?
>
> In response to your question, both [1] and [2] used the final return as the performance metric. We reported both maximal return and final return when they differed, and only the final return when they were nearly the same.
>
> **References:**
>
> [1] "Implementation Matters in Deep Policy Gradients: A Case Study on PPO and TRPO" by  Engstrom et al., ICLR, 2019;
>
> [2] "Deep Reinforcement Learning that Matters" by Henderson et al., AAAI, 2017.

---

> > ### Comment · Reviewer_ikDh · 2024-11-27
> >
> > Thank you for your response.
> >
> > Most of my concerns are resolved. My feedback remains positive.

---

> > > ### Author Response · Authors · 2024-11-27
> > > **Thanks**
> > >
> > > We are delighted that our response has addressed your concerns. Thank you for your thoughtful feedback and support of our work!

---

### Official Review · Reviewer_Wwkj · 2024-11-04

**Soundness:** 4
**Presentation:** 3
**Contribution:** 2
**Rating:** 5
**Confidence:** 4

**Summary:**

This paper investigates the role of the importance sampling term w.r.t. numerical instabilities in policy optimization algorithms. It takes a close look at TRPO and PPO, and shows that instabilities naturally arise when Gaussian policies’ actions are clipped to fit a bounded continuous action space. The authors argue that this type of instabilities is different (and more important) from existing sources of instabilities, and that typical algorithmic tricks like KL-penalties don’t prevent these instabilities.

Overall I found this paper mostly well written and easy to follow. My main concerns are about the soundness of the author’s argument and the significance of the work. I think the instabilities described by the authors happen in contrived scenarios, namely when the policy’s support lies mostly outside of the action space.

**Strengths:**

- The paper is well-written and easy to follow, even for an audience that has little background in policy gradient for continuous control. I especially appreciated that the notation is kept simple, that only relevant equations are introduced when needed, and that the flow of the paper was natural. The authors do a good job of holding the reader’s hand throughout the paper.
- I think the paper is sound. All derivations looked right (caveat: I didn’t check the Appendix) and the authors analyze many of the aspects of policy gradient algorithms.
- The topic investigated — namely, how to stabilize and scale up policy gradient algorithms — is very relevant to the ICLR community. Especially since tuning PPO remains painful and the analysis isn’t tied to a particular application domain (modulo the action space assumptions).

**Weaknesses:**

- My main concern stems from the contrived analysis. The authors point three cases in which the importance sampling ratio can become numerically unstable for Gaussian policies (l. 212):
    1. If the covariance of the reference policy has large eigenvalues.
    2. If the support of the policy mostly lies outside of the action space.
    3. If the policy changes rapidly with mini-batch updates.

    The authors state that 1) can be controlled (eg, entropy bonus). They argue the main issue stems from 2), but I think it is unlikely unless the policy’s initialization is pathological. A sensible initialization would be to set the mean of the policy at the center of the action space, and then it’s unclear how it would drift outside of the action space. Regarding 3), I think this is directly targetted by the KL constrained. This argument is not refuted by the analysis in Section 4, which mostly targets the sources of instabilities due to 2).

- A second concern is that the authors study potential fixes but none works. While this is not required (pointing out an unknown issue is valuable in itself), it would have strengthened the analysis if it led to a solution. In fact, I believe there is a well-known solution: for bounded action spaces, use the so called “Tanh-Gaussian” distribution as done in the Soft Actor-Critic and derivative works (Haarnoja et al, 2018). The authors mention and try the Tanh-Gaussian distribution; yet, they do not cite Haarnoja.
- In fact, I’m doubtful of some of the experimental results. SAC with the Tanh-Gaussian gets among the best results on the Mujoco tasks studied in this paper but the authors report negative results when trying this distribution with TRPO / PPO. Having tried this the Tanh-Gaussian distribution on continuous control Mujoco tasks and with PPO, my intuition tells me it should perform no worse than a Gaussian distribution if tuned properly.
- Finally, there are some more minor issues. I think the introduction, abstract, and title are slightly over-claiming: they don’t qualify the claims that the importance sampling instabilities only arise for continuous and **bounded** action spaces. This should be mentioned early, not on p. 3 (l. 137). Some of the figures could also use some work; eg, I don’t know what’s the difference between the plots in Fig. 2: is it with clipped / tanh policies? is it different tasks? why does Max Ratio monotonically increase in (b) and (c) but not in (a)? is max_t computed over all optimization steps or just the current one (if former, then monotonic increase is not surprising)? Another example, for Table 1 it would be good to include the reference results without any clipping or tanh.

**Questions:**

Please see the “Weaknesses” section above. In addition:

- Why would the policy distribution drift significantly outside the box defined by the action space?
- l. 357: Why would $\mu$ be outside the action space at initialization?
- Do the same issues appear on discrete action spaces?

---

> ### Author Response · Authors · 2024-11-18
> **Rebuttal, part 1**
>
> Thank you for the detailed feedback. Please see the following response for the concerns.
>
> Regarding the weaknesses:
>
> > My main concern stems from the contrived analysis.
>
> It is really a good question! Let us clarify these points:
>
> > The authors state that 1) can be controlled (eg, entropy bonus).
>
> When the covariance of the policy has small eigenvalues, the importance sampling ratio becomes ill-conditioned. This issue can be controlled by setting lower bounds on the standard deviations.
>
> > They argue the main issue stems from 2), but I think it is unlikely unless the policy’s initialization is pathological.
>
> As mentioned in (3), the policy distribution can shift outside the action space during training due to mini-batch updates, even if the policy network is well-initialized. Therefore, the initialization itself is less important compared to the shift caused during policy training. This is not a contrived scenario, but rather the general case during PPO training. This phenomenon has been reported in previous work ([1]), though it was not discussed in relation to numerical issues. We have revised the wording to clarify this point.
>
> > I think this is directly targetted by the KL constrained.
>
> In Section 4 (now Section 5), we showed that the KL divergence is not able to fully control the mean from leaving the action space, since the KL divergence measures the averaged distance between two distributions over the entire space, whereas the probability ratio can take large values at specific points. In Figure 2 (now Figure 3), we observe that while the KL distance remains small throughout, the maximum of the probability ratios explodes.
>
> > A second concern is that the authors study potential fixes but none works.
>
> Since we have observed that vanilla policy gradient is numerically more robust compared to PPO, we propose a new solution called "vanilla policy gradient with clipping." This method removes the importance sampling ratio from the PPO objective and applies the same clipping rule to the vanilla policy gradient objective. The algorithm combines the effectiveness of PPO with the robustness of VPG. Please refer to the global response and the revised draft for further details.
>
> > In fact, I believe there is a well-known solution: for bounded action spaces, use the so called “Tanh-Gaussian” distribution as done in the Soft Actor-Critic and derivative works (Haarnoja et al, 2018).
>
> This approach still suffers from the ill-conditioned nature of importance sampling. For instance, consider the modified probability density function in the one-dimensional case, given by $ \pi'(a|s) = \pi(u|s) (1 - \tanh^2(u))^{-1} $ where $u \sim \pi$ is a Gaussian random variable and $a = \tanh(u)$. Note that when the mean of $\pi$ is too large, it is likely that the random variable $u$ will take on extreme values, resulting in an excessively small value for the term $1 - \tanh^2(u)$ and causing numerical overflow in subsequent steps. Thanks for the comment and we have added this discussion to the revised draft.
>
> > In fact, I’m doubtful of some of the experimental results. SAC with the Tanh-Gaussian gets among the best results on the Mujoco tasks studied in this paper but the authors report negative results when trying this distribution with TRPO / PPO.
>
> In fact, the approach we attempted in the "Mean transformation" section is to directly bound the mean of the Gaussian distribution, rather than its output. Therefore, these are different approaches.
>
> > I don’t know what’s the difference between the plots in Fig. 2.
>
> In Figure 2 (now Figure 3), we empirically show that optimization constraints such as clipping and KL penalties are not sufficient to prevent the importance sampling ratio from exploding. The difference between these plots lies in the constraints applied to the policy training: (a) probability ratio clipping, (b) KL-penalty, and (c) no constraints. We have clarified this point in the revision, and thank you for pointing it out.
>
> > is max computed over all optimization steps or just the current one
>
> Just the current step.
>
> > for Table 1 it would be good to include the reference results without any clipping or tanh.
>
> Now we have included the reference results without any clipping or tanh in Table 1.

---

> ### Author Response · Authors · 2024-11-18
> **Rebuttal, part 2**
>
> For the questions:
>
> > Why would the policy distribution drift significantly outside the box defined by the action space?
>
> It is very likely that the policy distribution shifts significantly outside the action space, as PPO performs multiple policy updates in each epoch due to mini-batching. It has been reported that PPO training often leads to policies that generate boundary actions ([1]).
>
> > l. 357: Why would $\mu$ be outside the action space at initialization?
>
> In line 357, we mentioned that when the policy network is large, the variance of its output is also large at initialization, which implies that some actions may already be outside the action space. While this is not necessarily always the case, it is more likely to occur with larger networks.
>
> > Do the same issues appear on discrete action spaces?
>
> That’s a good question. Although it is somewhat beyond the scope of this work, we believe that numerical instability could also arise in discrete action spaces. The root causes of instability—policy distribution shifts and singular covariance matrices—are not restricted to the continuous case.
>
> We hope that our response can address your concerns, and we are happy to answer any further questions.
>
> **References:**
>
> [1] Is Bang-Bang Control All You Need? Solving Continuous Control with Bernoulli Policies, Seyde et al. NeurIPS, 2021.

---

> > ### Comment · Reviewer_Wwkj · 2024-11-25
> >
> > Thank you for your detailed response. Several of my concerns have been addressed, I only have two that remain.
> >
> > First, I'm still unconvinced that Tanh-Gaussian w/ PPO / TRPO doesn't work. I understand that if $u$ is large, $\pi(a \mid s)$ is ill-conditioned but I don't see how $u$ could become so large — this gets to my second point.
> >
> > Second, and more importantly, my main concern remains: namely, I do not see how the optimized policy can drift so as to cause instabilities in the importance sampling ratio. It seems unlikely because the importance ratio is between a policy and the same policy from a couple of mini-batches ago. So it's hard to believe they'll be very different and drift outside the action space box. You state "This is not a contrived scenario, but rather the general case during PPO training." — could you give a compelling argument or an illustrative example to help convince me?

---

> > > ### Author Response · Authors · 2024-11-25
> > > **Reply to Reviewer Wwkj**
> > >
> > > Thank you for your comment. We will address your question in three steps:
> > >
> > > > Is it possible that the mean of the Gaussian policy drifts outside the action space in practice?
> > >
> > > We have added an illustrative experimental result (Figure 1(c)) showing that the action with the largest norm in each mini-batch grows outside the boundary during PPO training in the Humanoid-v4 environment. Additionally, the experimental result shown in Figure 3 of [1] indicates that the means of a Gaussian policy trained by PPO tend to leave the action box, thereby generating boundary actions. A similar observation was also made in [2]. These findings provide concrete evidence supporting our claim: "This is not a contrived scenario, but rather the general case during PPO training."
> > >
> > > > Why does the policy tend to drift outside the boundaries during PPO training?
> > >
> > > Generally speaking, PPO tends to find policies that generate boundary actions because this behavior aligns with what an optimal policy does. In other words, optimal policies are usually bang-bang controllers in most continuous-control problems. A thorough theoretical analysis of this phenomenon can be found in Section 3 of [1].
> > >
> > > > How does a large mean cause numerical issues in importance sampling?
> > >
> > > This is one of the core results presented in Section 3, which explains that the growth rate of the importance sampling ratio is significantly influenced by the distance between the mean, $\mu(s; \theta)$, and the clipped/transformed action, $\phi(a)$, as described in Equation 6 of the draft. Consequently, when the mean of the Gaussian policy drifts outside the action box, the probability ratio can grow exponentially large during mini-batch training. To support this observation, we have included experimental results in Figure 2, which demonstrate how the probability ratio becomes ill-conditioned in practical PPO training.
> > >
> > > **References:**
> > >
> > > [1] Is Bang-Bang Control All You Need? Solving Continuous Control with Bernoulli Policies, Seyde et al., NeurIPS, 2021;
> > >
> > > [2] Remember and Forget for Experience Replay, Novati et al., ICML 2019.

---

> ### Author Response · Authors · 2024-12-01
>
> Dear Reviewer Wwkj,
>
> Thank you once again for your constructive comments. As the discussion period is closing soon, we hope our response has addressed your main concerns. Please let us know if you have any remaining questions.

---

### Official Review · Reviewer_7dTj · 2024-11-04

**Soundness:** 3
**Presentation:** 3
**Contribution:** 3
**Rating:** 6
**Confidence:** 3

**Summary:**

This paper investigates numerical instabilities in policy gradient methods used in deep reinforcement learning. It focuses on the gradient explosion issues when training deep policy gradient algorithms like TRPO and PPO.  The authors argue that these instabilities are often due to the ill-conditioned density ratio in importance sampling used in algorithms like TRPO and PPO which will lead to large gradient magnitudes that can exceed the limits of floating-point representation. The authors demonstrate that existing constraints like KL divergence and probability ratio clipping are insufficient to address these numerical issues. Several techniques and approaches are proposed to mitigate these instabilities.

**Strengths:**

- The paper provides a theoretical analysis of the cause of numerical instabilities in deep policy gradient algorithms. The paper identifies the role of importance sampling and density ratios in gradient explosion araising from numerical instabilities.
- This paper critiques the limitations of existing optimization constraints such as the KL divergence and ratio clipping in maintaining numerical stability.
- This paper provides some potential solutions to mitigate the numerical instabilities in deep policy gradient algorithms.

**Weaknesses:**

- The paper does not provide a detailed empirical analysis of the numerical instabilities to support the theoretical claims.
- The propoesed solutions are either difficult to implement or hurt the performance of the algorithms. There is no clear and effective solution presented in this paper.
- Although the paper provides some experimental results for the proposed results, they are not comprehensive and do not provide a clear comparison with existing methods.

**Questions:**

- Can you provide more insights into the numerical instabilities in policy gradient methods and how they manifest during training? How do these instabilities affect the convergence and performance of deep reinforcement learning algorithms?
- Can you provide a comprehensive empirical study on the proposed solutions to mitigate numerical instabilities in policy gradient methods? Can you explain more about the limitations of these solutions?

---

> ### Author Response · Authors · 2024-11-18
> **Rebuttal**
>
> Thank you for your important feedback. We hope that the following responses may address the questions and concerns.
>
> Regarding the weaknesses:
>
> > The paper does not provide a detailed empirical analysis of the numerical instabilities to support the theoretical claims.
>
> We have conducted an empirical analysis of the influence of optimization constraints and the impact of large means on numerical stability. Some experimental results were initially placed in the Appendix due to the page limit. We have now added a new section in the main text dedicated to the empirical analysis. Please refer to the revised draft for details.
>
> > The propoesed solutions are either difficult to implement or hurt the performance of the algorithms. There is no clear and effective solution presented in this paper.
>
> Since we observed that the vanilla policy gradient (VPG) is numerically more robust compared to PPO, we propose a new solution called "vanilla policy gradient with clipping." This approach simplifies the PPO objective by removing the importance sampling ratio while applying the same clipping rule to the VPG objective. The resulting algorithm combines the effectiveness of PPO with the robustness of VPG. Please refer to the global response and the revised draft for further details.
>
> > Although the paper provides some experimental results for the proposed results, they are not comprehensive and do not provide a clear comparison with existing methods.
>
> We have expanded the experimental results in the main text; please refer to the revised draft for details.
>
> Regarding the questions:
>
> > Can you provide more insights into the numerical instabilities in policy gradient methods and how they manifest during training? How do these instabilities affect the convergence and performance of deep reinforcement learning algorithms?
>
> Based on our analysis, the numerical instability of TRPO/PPO arises from large means and/or small standard deviations in the stochastic policy during training. When such instabilities occur, the algorithm may encounter arithmetic overflow (e.g., division by zero), leading to NaN or Inf values. While this issue has been recognized as a fundamental challenge in policy gradient methods (as discussed in https://stable-baselines3.readthedocs.io/en/master/guide/checking_nan.html), it is often regarded as a hyperparameter tuning problem. However, the root cause has not been well understood.
>
> > Can you provide a comprehensive empirical study on the proposed solutions to mitigate numerical instabilities in policy gradient methods? Can you explain more about the limitations of these solutions?
>
> We have proposed a solution that effectively addresses the numerical issues caused by importance sampling. Please refer to the revised draft for further details.
>
> Please let us know if you have any further questions.

---

> > ### Author Response · Authors · 2024-11-28
> >
> > Dear Reviewer 7dTj,
> >
> > Thank you once again for your time and effort in reviewing our work! We hope our response has addressed your concerns. As the discussion period comes to a close, please let us know if you have any further questions.

---

> > > ### Comment · Reviewer_7dTj · 2024-12-02
> > >
> > > Thank you for your response. My concerns are addressed. I will raise my score to be positive.

---

> > > > ### Author Response · Authors · 2024-12-02
> > > >
> > > > Dear Reviewer 7dTj,
> > > >
> > > > We sincerely thank you for supporting our work and raising the score!

---

### Author Response · Authors · 2024-11-18
**Global response**

We would like to thank the reviewers for their constructive feedback! Among the comments, we found that most reviewers expressed concerns about the proposed solutions to overcome the numerical issues. In response, we have revised the draft by adding a new solution, called "vanilla policy gradient with clipping," at the end of Section 5. This algorithm simply removes the importance sampling ratio from the PPO objective and applies the same clipping rule to the vanilla policy gradient objective, which has been shown to be numerically stable. The resulting algorithm combines the effectiveness of PPO with the robustness of VPG, and the objective is

$$L^{CPG}(\theta) = \hat{\mathbb{E}}_{(s_t, a_t) \sim \pi} [min ( \log p \hat{A}^{\pi}_t, clip(\log p, \log (1-\epsilon), \log (1+\epsilon)) \hat{A}^{\pi}_t ))].$$

We found that it can achieve performance similar to PPO without any numerical issues. Additionally, we have updated the main text to reflect this modification.

A summary to the major changes in the revision:

* In Section 3, we added a paragraph explaining why the tanh-Gaussian distribution also suffers from policy distribution shift.

* We added a section after Section 3 to present empirical evidence that supports our theoretical analysis. We found that the experimental results related to mean transformation provide stronger evidence for our analysis than simply being a potential approach, so we have moved it to this section.

* We relegated some possible approaches to the Appendix, as they are less effective than the one we have proposed but may still serve as useful references for the audience.

We also provide a summary to the contributions of the revised work:

* We provide a theoretical analysis of how ill-conditioned importance sampling ratios lead to numerical issues in PPO, supported by empirical evidence.

* We also demonstrate that optimization constraints, such as ratio clipping and KL-penalty, are not sufficient to prevent these issues.

* Motivated by vanilla policy gradient, we propose an algorithm that makes a slight modification to PPO, effectively preventing gradient explosion while achieving performance comparable to PPO.

---

### Meta-Review · Area_Chair_Qgne · 2024-12-21

**Metareview:**

This paper examines the performance of PPO/TRPO and investigates the numerical instability during training caused by the importance sampling ratio. Using examples, the authors demonstrate that numerical instability in importance sampling can lead to gradient explosion and provide several potential solutions, though none work perfectly. While importance sampling and its extreme values in long-horizon tasks have been studied in the literature, the focus on numerical instability is novel, with the authors claiming that this issue can arise even during a single evaluation.

The reviewers expressed strong interest in the results presented in the paper. Some questions were raised, but the authors addressed these by providing additional empirical results and external references to support their observations. Despite this, some reviewers still feel the paper is somewhat overclaiming due to the limited experiments conducted in just three MuJoCo environments, which may not be sufficient to draw statistically significant conclusions. Another minor limitation is that the analysis applies only to Gaussian policies. Notably, while the authors presented several solutions, none effectively resolved the issue in the original paper.

During the discussion phase, the authors introduced a new solution called “vanilla policy gradient with clipping,” which was not included in the original submission and was not thoroughly evaluated by all reviewers. More experiments would be needed to verify this new approach, and the paper may require another round of review.

Personally, I find this paper to be very interesting and potentially impactful for the community. However, given the current limitations, I recommend a weak rejection and look forward to a more polished version of this promising work.

**Additional Comments On Reviewer Discussion:**

The reviewers identified three critical gaps: the absence of a working solution, insufficient statistical evidence for the results, and limited analysis across policy classes. The authors' response does not adequately address these fundamental concerns.

---

### Decision · Program_Chairs · 2025-01-22

Reject